# Multivariate cross-frequency coupling via generalized eigendecomposition

Michael X Cohen*

Donders Center for Neuroscience, Radboud University Nijmegen Medical Centre, Radboud University, Nijmegen, Netherlands

**Abstract** This paper presents a new framework for analyzing cross-frequency coupling in multichannel electrophysiological recordings. The generalized eigendecomposition-based cross-frequency coupling framework (gedCFC) is inspired by source-separation algorithms combined with dynamics of mesoscopic neurophysiological processes. It is unaffected by factors that confound traditional CFC methods—such as non-stationarities, non-sinusoidality, and non-uniform phase angle distributions—attractive properties considering that brain activity is neither stationary nor perfectly sinusoidal. The gedCFC framework opens new opportunities for conceptualizing CFC as network interactions with diverse spatial/topographical distributions. Five specific methods within the gedCFC framework are detailed, these are validated in simulated data and applied in several empirical datasets. gedCFC accurately recovers physiologically plausible CFC patterns embedded in noise that causes traditional CFC methods to perform poorly. The paper also demonstrates that spike-field coherence in multichannel local field potential data can be analyzed using the gedCFC framework, which provides significant advantages over traditional spike-field coherence analyses. Null-hypothesis testing is also discussed.

## Introduction

**\*For correspondence:**
mikexcohen@gmail.com

**Competing interests:** The author declares that no competing interests exist.

Cross-frequency coupling (CFC) refers to the phenomenon in which dynamics across two or more frequency bands are related. CFC has been reported in many cortical and subcortical brain regions in multiple species, and has inspired computational models and theories (*Canolty and Knight, 2010*; *Fell and Axmacher, 2011*; *Lisman and Jensen, 2013*). A core idea that permeates CFC theories is that the phase of a slower brain rhythm coordinates a temporal sequence of faster processes that represent specific items in memory or in sensory space. There are several manifestations of CFC (*Canolty and Knight, 2010*; *Hyafil et al., 2015*; *Jensen et al., 2007*; *Jirsa and Müller, 2013*), and the most widely used CFC analysis methods involve the application of Euler's formula, phase synchronization, or distribution analyses, with the goal of determining whether high-frequency power values are non-uniformly distributed over low-frequency phase (*Tort et al., 2010*).

Nevertheless, the standard corpus of CFC measures is increasingly criticized, with valid concerns about spurious or biased CFC estimates resulting from non-stationarities, sharp transients, non-uniform phase distributions, and other issues (*Aru et al., 2015*; *Kramer et al., 2008*; *Lozano-Soldevilla et al., 2016*; *van Driel et al., 2015*). These potential biases are not fatal flaws, and in some cases can be alleviated by using alternative methods, shuffling-based permutation testing, or debiasing terms (*Cohen, 2014*; *Pittman-Polletta et al., 2014*; *van Driel et al., 2015*; *Voytek et al., 2013*). However, these concerns limit the ubiquitous applicability of standard CFC measures.

The source of these potential biases comes from violating the assumption that brain oscillations are sinusoidal with a stable frequency. This assumption stems from the use of sinusoidal narrowband filters, such as Morlet wavelets or narrow FIR filters. On the other hand, it is increasingly becoming clear that neural oscillations are not always sinusoidal (*Jones, 2016*; *Mazaheri and Jensen, 2008*).

The non-sinusoidal waveform shape of neural oscillations may provide important insights into underlying biophysical processes, and yet, for CFC analyses, the non-sinusoidal waveform shapes are potential confounds that must be avoided or corrected. Therefore, analysis methods that do not assume sinusoidality are important 1) for methodological reasons to examine CFC in a wider range of datasets and 2) for theoretical reasons to facilitate discovery of CFC regardless of waveform shape (or explicitly incorporating waveform shape), which is an important step towards uncovering the neurobiological origins and functional significances of neural oscillations.

Multichannel recordings provide new opportunities for the discovery of physiologically interpretable patterns of CFC. For example, theories of CFC suggest that the low-frequency rhythm regulates the timing of different networks (*Lisman and Jensen, 2013*), implying that the neural circuits that produce the low- and high-frequency rhythms should have different electrical projections onto multi-electrode recordings. Furthermore, because volume conduction carries activity from a single neuro-electric source to multiple electrodes, incorporating weighted averages from multiple electrodes can boost signal-to-noise ratio and help in isolating patterns of activity (*Parra et al., 2005*), including in CFC analyses (*Canolty et al., 2012*; *Soto et al., 2016*). This is particularly relevant for noninvasive recordings, in which high-frequency activity can have low power and can be contaminated by muscle artifacts. Finally, multichannel recordings allow the application of advanced matrix analysis methods, including dimensionality-reduction and source separation algorithms, which can identify spatiotemporal patterns of activity that might be difficult to uncover when considering activity from a single electrode at a time (*Grootswagers et al., 2016*; *Orekhova et al., 2011*).

The purpose of this paper is to present a new hypothesis-driven framework for conceptualizing and quantifying CFC in multichannel datasets. This framework is based on using generalized eigen-decomposition (GED) of multichannel covariance matrices, and is therefore termed generalized eigendecomposition-based cross-frequency coupling (gedCFC). The backbone of gedCFC for source separation and dimensionality reduction is grounded in decades of mathematics and problem-solving in engineering, with applications in neuroscience (*Parra et al., 2005*; *Särelä and Valpola, 2005*). This paper will show that gedCFC has several advantages over the commonly used CFC methods, including increasing signal-to-noise characteristics and avoiding confounds of non-sinusoidal oscillations. Perhaps the most important advantages, however, are its potential for use in hypothesis-driven network discovery and its increased flexibility and robustness for uncovering CFC patterns in nonstationary multichannel data.

Five specific methods are derived from this framework (a brief overview is provided in *Table 1* for reference), with each method designed for different assumptions about underlying neural processes and different analysis goals. Each method is validated in simulated data, and proof-of-principle applications are shown in empirical data recorded in humans (EEG, MEG, ECoG) and in rodents (LFP and spike-LFP). Furthermore, it is also demonstrated that spike-field coherence in multichannel datasets can be conceptualized as a special case of CFC, permitting the application of gedCFC in this

**Table 1.** Overview of methods.

| | Description | Analysis goal | Key assumption |
|---|---|---|---|
| Method 1 | **S** defined by peri-LF-peak; **R** defined by all data | Identify a single phase-amplitude coupled network | One HF network with power proportional to LF phase |
| Method 2 | **S** defined by peri-LF-peak; **R** defined by peri-LF-trough | Identify two networks that alternate according to LF phase. | Two different HF networks that have power peaks at different LF phases |
| Method 3 | LF activity bias-filters sphered data | Use (possibly nonstationary) LF waveform shape to identify a HF component. | Well-defined LF waveform |
| Method 4 | Delay-embedded matrix, **S** and **R** defined as in Methods 1 or 2 | Empirically determine a CFC-related spatiotemporal filter | Appropriate delay-embedded order |
| Method 5 | Similar to Method 4 but data are taken peri-action potential | Empirically determine a spatiotemporal LFP filter surrounding action potentials | Sufficient delay-embedding order; one peri-spike network |

*Notes.* LF = low frequency; HF = high frequency. All methods make the assumption that the spatiotemporal characteristics of the HF activity are stable over repeated time windows from which covariance matrices are computed.

context. Advantages, assumptions, and limitations underlying the different methods are discussed, along with suggestions for practical aspects of data analysis and inferential statistics.

## Generalized eigendecomposition

This paper describes the gedCFC framework in a manner that is approachable to scientists with diverse backgrounds; deeper mathematical discussions that justify using GED to optimize linear spatiotemporal filters according to user-specified objectives are presented in many other publications (*Blankertz et al., 2008*; *de Cheveigné and Parra, 2014*; *Nikulin et al., 2011*; *Parra et al., 2005*; *Tomé, 2006*).

The goal of gedCFC is to create a component, formed from a weighted sum of all electrodes, that optimizes the ratio between user-specified minimization and maximization criteria. For example, a component might maximize the difference between high-frequency activity that appears during peaks of a low-frequency oscillation, versus high-frequency activity that is unrelated to the low-frequency oscillation. This component is created through a matrix decomposition procedure called eigendecomposition.

Eigendecomposition (also called eigenvalue or eigenvector decomposition) involves finding certain vectors that are associated with square matrices. The basic eigenvalue equation is $\mathbf{Sw} = \mathbf{w}\lambda$, where $\mathbf{S}$ is a square matrix, $\mathbf{w}$ is a vector, and $\lambda$ is a single number called an eigenvalue. This equation means that multiplying the eigenvector $\mathbf{w}$ by matrix $\mathbf{S}$ has the same effect as multiplying $\mathbf{w}$ by a single number $\lambda$. In other words, matrix $\mathbf{S}$ does not change the direction of $\mathbf{w}$, $\mathbf{S}$ merely stretches or shrinks it. If $\mathbf{S}$ is a channel-by-channel covariance matrix formed by multiplying the M-by-N (channels by time points) data matrix by its transpose, then the eigenvector $\mathbf{w}$ points in the direction of maximal covariance. The full set of eigenvectors produces matrix $\mathbf{W}$ (with corresponding eigenvalues in the diagonal matrix $\mathbf{\Lambda}$) that spans the space of $\mathbf{S}$ using basis vectors that each account for maximal variance in $\mathbf{S}$ while selected to be pairwise orthogonal. This is known as a principal components analysis.

The eigenvalue equation is generalized to two square matrices $\mathbf{R}$ and $\mathbf{S}$, as $\mathbf{SW} = \mathbf{RW\Lambda}$. This equation can be more intuitively understood by moving $\mathbf{R}$ to the left side via multiplication of its inverse: $(\mathbf{R}^{-1}\mathbf{S})\mathbf{W} = \mathbf{W\Lambda}$. $\mathbf{R}^{-1}\mathbf{S}$ is the matrix analog of $\mathbf{S}$ divided by $\mathbf{R}$ (think of $\frac{2}{3}$ as being equivalent to $3^{-1}\times 2$). If $\mathbf{S}$ is a covariance matrix of a 'signal' dataset and $\mathbf{R}$ is a covariance matrix of a 'reference' dataset, then GED can be understood to produce eigenvectors that identify directions of maximal power ratio (highest gain) in the matrix product $\mathbf{R}^{-1}\mathbf{S}$, in other words, directions that best differentiate matrices $\mathbf{S}$ from $\mathbf{R}$. The columns of $\mathbf{W}$ are called spatial filters (spatiotemporal filters for delay-embedded matrices) or unmixing coefficients, and the column with the largest associated eigenvalue is the one that maximally differentiates the two matrices. The filters can be applied to the data to compute components, and they can be visualized by inspecting the columns of $\mathbf{W}^{-\mathbf{T}}$, which is also called the forward model of the filter or the activation pattern (*Haufe et al., 2014*).

gedCFC identifies multichannel CFC-related networks by contrasting covariance matrices computed from to-be-maximized data features (matrix $\mathbf{S}$) against to-be-minimized data features (matrix $\mathbf{R}$). The two covariance matrices should be similar enough to suppress CFC-unrelated activity, while being different enough to isolate the neural networks that exhibit CFC. For example, one could identify a theta-band (~6 Hz) rhythm and then compute matrix $\mathbf{S}$ from peri-peak data and matrix $\mathbf{R}$ from peri-trough data; the GED of $\mathbf{S}$ and $\mathbf{R}$ would reveal a component that maximally differentiates peak-related from trough-related activity, which is a manifestation of network-level phase-amplitude coupling.

The spatial filter comprises values for each electrode that are used to compute a weighted sum of activity from all electrodes. The resulting time series is the gedCFC component. Note that although defining the spatial filter involves two covariance matrices that might be computed from discrete time windows, the gedCFC component is a "continuous" signal to which time-domain, frequency-domain, or time-frequency domain analyses can be applied.

In addition to being able to specify both maximization and minimization criteria, another important distinction between GED of two covariance matrices and eigendecomposition of one matrix is that eigenvectors are selected to form an orthogonal set only when the matrix $\mathbf{S}$ is symmetric. Although both $\mathbf{S}$ and $\mathbf{R}$ are symmetric covariance matrices, $\mathbf{R}^{-1}\mathbf{S}$ is generally not symmetric. In other words, the eigenvectors are independent but not orthogonal. This is an important advantage over

principal components analysis, which often performs poorly when used as a source separation method (*Delorme et al., 2012*) (see also *Figure 1—figure supplement 1*).

## Using GED to identify the low-frequency rhythm

The GED framework is not limited to CFC; it is also recommended that GED is used to define a spatial filter that maximizes power at the lower frequency (*Blankertz et al., 2008*; *Nikulin et al., 2011*; *Parra and Sajda, 2003*; *Särelä and Valpola, 2005*). There are several ways to define the **S** and **R** covariance matrices (see *Cohen, 2016*, for comparisons); the approach taken here is to compute **S** from narrow-band filtered data and **R** from the broadband data (*de Cheveigné and Arzounian, 2015*). The eigenvector with the largest eigenvalue is taken as a spatial filter optimized for activity in that frequency band. The advantages of GED for source separation of narrowband activity include increased signal-to-noise ratio, more accurate recovery of source time course dynamics, and eliminated necessity of electrode selection.

The MATLAB code used to simulate data and to implement the five methods presented here are available online (mikexcohen.com/data). Readers are encouraged to reproduce the findings, modify the code to evaluate success and failure scenarios, and adapt and extend the code to specific data analysis applications.

## Results

### Method 1: gedCFC on trough-locked covariance vs. entire time series covariance

Method 1 is designed for a situation in which the activity of one network fluctuates as a function of the phase of a lower-frequency network. To implement Method 1, a component (or single electrode) from which to compute the lower-frequency rhythm is first defined and then the troughs are identified (or peaks, or rising slopes, or any theoretically relevant phase position). The **S** covariance matrix is computed from data surrounding each trough, and the **R** covariance matrix is computed from the entire time series. *Figure 1—figure supplement 2* provides a graphical overview of Method 1.

Simulated EEG data to illustrate this method were constructed by generating activity in three dipoles in occipital cortex: one to provide a theta rhythm, and two to provide gamma oscillations. The topographical projections of these dipoles were overlapping but not identical (*Figure 1a*). The theta rhythm (peak frequency of 6 Hz) contained amplitude and frequency non-stationarities, one gamma dipole had a peak frequency at 40 Hz and an amplitude modulated by theta phase, and the other gamma dipole had a peak frequency at 50 Hz and an amplitude that was modulated independently of theta phase. This second dipole served as a 'distractor' to test whether it would produce an artifact. These dipole time series, along with random correlated 1/f noise at 2,001 additional brain dipoles, were projected to 64 scalp EEG channels (see Materials and methods for additional details about simulations).

GED was used to extract a component that maximized power in the theta band. The forward model of the filter closely matched the dipole projection (*Figure 1a*), indicating accurate reconstruction of the source activity. The **S** covariance matrix was formed using broadband data from ¼ of a theta-frequency cycle surrounding each theta trough (⅛ of a cycle before and ⅛ after the trough), and the **R** covariance matrix was formed using the entire broadband time series. The MATLAB code used to obtain the weights is `[V,D]=eig(covTrough, covTot)`. The eigenvector in matrix V with the largest eigenvalue (diagonals of matrix D) is the spatial filter that can be used to obtain the 'trough component'. This component is a weighted sum of all electrodes that maximally differentiates activity during theta troughs from activity during all theta phases.

Method 1 accurately reconstructed both the topographical map and the relationship between gamma power and theta phase (*Figure 1*). It is noteworthy that the channel data were not bandpass filtered; the 40 Hz gamma component was identified because of the separability of the covariance matrices. It is also noteworthy that the 50 Hz gamma component had twice the power of the 40 Hz gamma component (*Figure 1b*) and yet was fully suppressed by the spatial filter. This occurred because its dynamics were uncorrelated with theta phase, meaning that activity from the 50 Hz dipole contributed equally to both **S** and **R** covariance matrices.

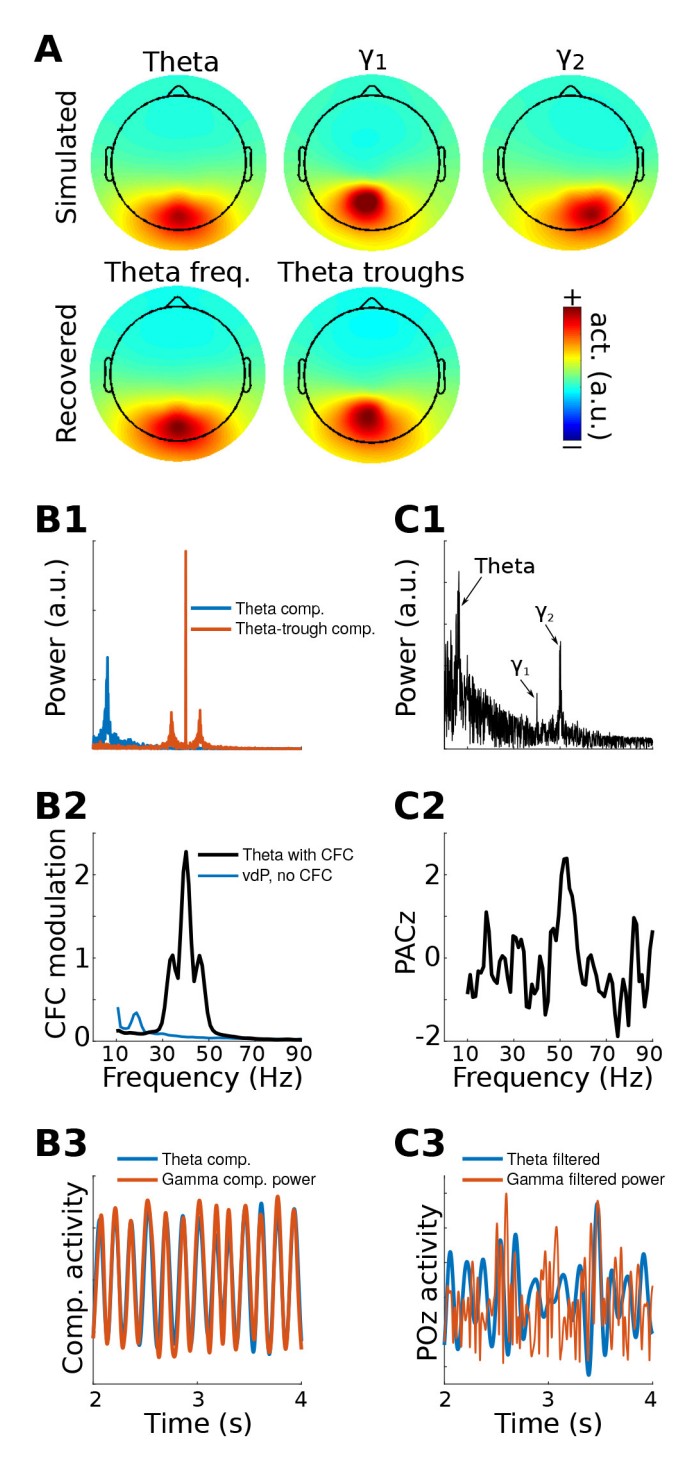

**Figure 1.** gedCFC Method 1 applied to simulated data. Data were generated in three dipoles in the brain and projected to the scalp (top row of (**A**)) along with correlated 1/f noise at 2,001 other dipoles. The theta component (~6 Hz) was recovered by GED (bottom row of (**A**)). The 40 Hz component was recovered using gedCFC on broadband data. (**B**) shows the power spectrum of the theta and theta-trough components (**B1**), the strength of CFC (**B2**), and illustrative time courses (**B3**). The blue line in (**B2**) corresponds to a van der Pol oscillator (vdP), which causes spurious high-frequency CFC in traditional analyses. (**C1**) shows the power spectrum of electrode POz, which is the spatial maximum of the EEG data power. Note that the 50 Hz gamma (γ2) has larger power than the 40 Hz (γ1). (**C2**) illustrates that traditional phase-amplitude coupling with permutation testing (PACz) applied to
*Figure 1 continued on next page*

*Figure 1 continued*

channel POz failed to recover the simulated CFC patterns. (**C3**) shows the poor reconstruction of the dynamics when using data from POz (compare with (**B3**)).

The following figure supplements are available for figure 1:

**Figure supplement 1.** Two-dimensional example of using generalized eigendecomposition (GED) to separate sources.

**Figure supplement 2.** Graphical overview of Method 1.

**Figure supplement 3.** This figure is similar to *Figure 1*, except that the gamma modulation was simulated in a collection of electrodes instead of in a dipole projected to the scalp.

CFC strength can be quantified in a number of ways. One approach is to apply narrowband temporal filters to the theta-trough component and to compute the average peak minus trough amplitude differences. This procedure revealed a maximum at 40 Hz and smaller peaks at 40 ± 6 Hz, corresponding to the nonstationarities induced from the theta rhythmicity of the 40 Hz amplitude modulations (*Figure 1b*). Other quantitative methods, including computing time-frequency power spectra surrounding troughs, $R^2$ fits between the low-frequency time series and the CFC component power time series, and phase synchronization between the two components, will be illustrated in later sections. Time courses of the reconstructed theta component and power time series of the theta-trough component revealed strong co-rhythmicity, as was constructed in the simulation (*Figure 1b*).

A traditional phase-amplitude coupling analysis based on Euler's formula from POz (the electrode with maximum power) failed to capture the simulated pattern of CFC due to low signal-to-noise characteristics of the simulation. Furthermore, the identical temporal filters applied to electrode POz showed little suggestion of CFC (*Figure 1c*). Together, these results show that even in simple simulations, gedCFC can accurately identify weak patterns of CFC in noisy data while traditional CFC methods can produce uninspiring results.

To demonstrate that gedCFC is robust to topographical distribution, this simulation was repeated by having several different electrodes (not dipoles projected to electrodes) that formed a network of theta phase-coupled gamma oscillators. The neurophysiological interpretation of this simulation is that a single theta generator regulates the timing of a spatially distributed synchronous gamma-oscillating network. The results remained robust, as seen in *Figure 1—figure supplement 3*. Although this is a useful demonstration, it is not a surprising finding, considering that gedCFC makes no assumptions about topographical characteristics or spatial smoothing.

Non-sinusoidal oscillations produce spurious phase-amplitude coupling, either due to non-uniform distribution of phase angles or to fast derivatives causing high-frequency spikes at certain phases (*Kramer et al., 2008*; *van Driel et al., 2015*). Therefore, this simulation was repeated, replacing the theta sine wave with repeated Gaussians or van der Pol oscillators at theta frequency (these two waveforms are known to produce spurious phase-amplitude coupling). gedCFC was completely robust to these non-sinusoidal time series, in that (1) the simulated pattern of CFC was accurately recovered regardless of non-stationarities or a non-sinusoidal shape of the lower frequency rhythm, and (2) when gamma oscillations were not amplitude-modulated by the lower frequency rhythm, no spurious CFC was identified. The latter is illustrated in *Figure 1b*, and additional simulations are shown in the online MATLAB code.

As a proof-of-principle application, Method 1 was applied to resting-state MEG data taken from the Human Connectome Project (*Van Essen et al., 2013*). The first step was to identify an alpha component by comparing covariance matrices from data filtered in the alpha range (peak 11 Hz; FWHM 5 Hz; these parameters were selected on the basis of visual inspection of power spectra from posterior sensors). There were several physiologically plausible alpha components, as expected on the basis of previous work (*Haegens et al., 2014*; *van der Meij et al., 2016*; *Walsh, 1958*). The alpha component with the largest eigenvalue was selected for subsequent analyses; it is possible that additional cross-frequency coupling dynamics could emerge when examining other alpha

components. A covariance matrix of broadband data was constructed around alpha peaks (only peaks corresponding to >1 standard deviation above the mean alpha peak amplitudes were included; this limits the analysis to periods of high alpha power) and compared against the broadband covariance from the entire resting period. The three gedCFC components with the largest eigenvalues were selected for analysis and visualization. Results show that, in this dataset, bursts of broadband and high-frequency activity were time-locked to alpha phase, ranging up to 150 Hz for components 2 and 3 (*Figure 2*), which appears consistent with previous reports (*Osipova et al., 2008*). Note that the different topographies of the alpha and alpha-peak components rule out the possibility that the cross-frequency coupling resulted from an artifact of the non-sinusoidal shape of alpha.

The primary assumptions of Method 1 are that there is only one high-frequency network of interest, and that its spatiotemporal characteristics (and therefore its covariance matrix) are consistent across each trough. The lower-frequency rhythm is assumed to be sufficiently rhythmic to be able to identify peaks and troughs, although frequency stationarity is not required; it would be valid to use, for example, empirical mode decomposition (a time-frequency decomposition method that can estimate non-sinusoidal oscillations) to obtain the lower-frequency rhythm. The primary limitation of Method 1 is that it is valid only when the single-network assumption is valid.

## Method 2: gedCFC on covariances of peri-peak vs. peri-trough

Method 2 is an extension of Method 1, and is designed for a situation in which a low-frequency rhythm regulates the timing of *two different* networks that are activated at distinct low-frequency

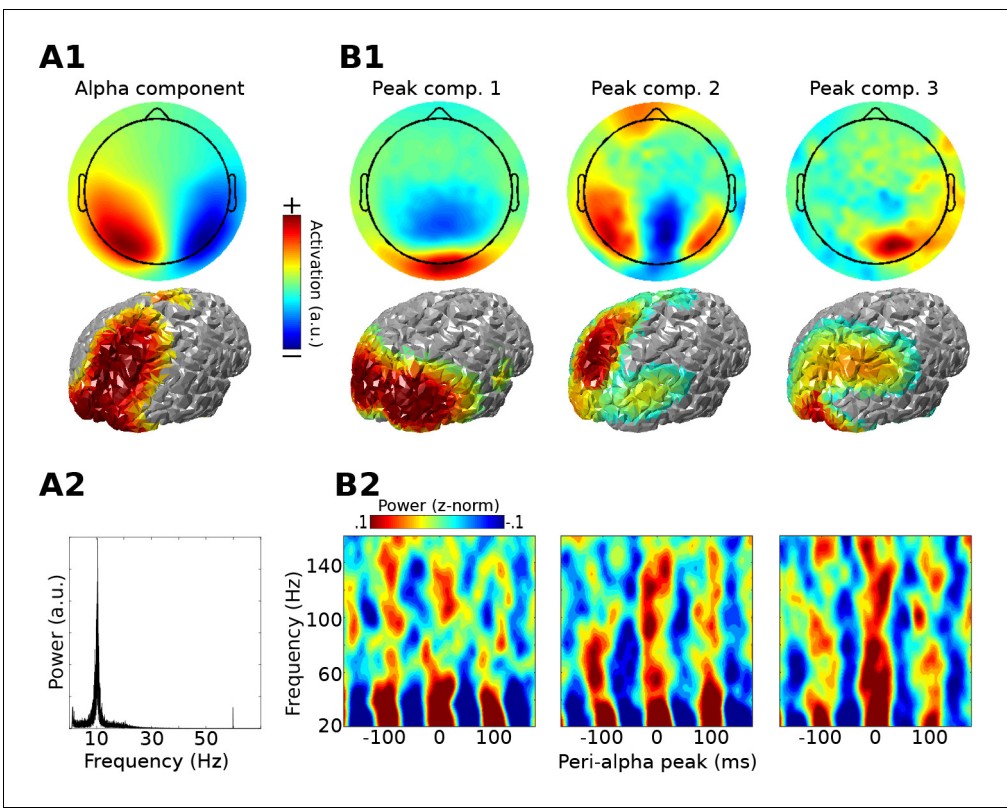

**Figure 2.** Method 1 applied to empirical MEG resting-state data. An alpha component was computed using GED; its topographical projection and power spectrum are shown in (**A1**) and (**A2**). Covariance matrices from broadband data surrounding high-power alpha peaks were computed, and were entered into gedCFC using a covariance matrix from the broadband signal from the entire time series as reference. The first three components (topographical projections and projections onto the cortical surface in (**B1**)) along with their time-frequency power spectra locked to alpha peaks (**B2**) are shown.

phases. This might be the case, for example, if theta phase controls the timing of different populations of neurons that represent different items held in memory (*Heusser et al., 2016*; *Zheng et al., 2016*).

The implementation is similar to that of Method 1, except that the two covariance matrices are formed from data around the trough and around the peak (or any other two phase regions). The MATLAB code is `[V,D]=eig(covTrough, covPeak)`. The eigenvector in matrix V with the largest eigenvalue (diagonals of matrix D) is the spatial filter used to obtain the 'trough component,' while the eigenvector with the smallest eigenvalue is the spatial filter used to obtain the 'peak component.' *Figure 3—figure supplement 1* provides a graphical overview of Method 2.

EEG data were simulated as for Method 1, except that activity from one dipole had a peak frequency at 40 Hz and an amplitude modulated by theta phase, while activity from the second dipole had a peak frequency at 45 Hz and an amplitude modulated by the inverse of theta phase. However, different frequencies are not necessary, and the two dipoles could oscillate at the same frequency. The key to their separability is topographical projections that produce differentiable covariance matrices, and different amplitude time courses with respect to the lower-frequency phase.

gedCFC recovered the cross-frequency dynamics of the network, accurately identifying the topographical distributions, frequency ranges, and time courses (*Figure 3*). Time-frequency power spectra time-locked to the theta troughs and peaks accurately captured the simulated dynamics. Note that this result emerged despite the absence of visually prominent gamma peaks in the channel power spectra. In other words, consistent spatiotemporal patterns that allow component extraction and CFC identification do not necessarily require visually compelling spectral peaks, as has been suggested by *Aru et al. (2015)*; (see *He et al. (2010)* for another counter-example). This is because the power spectrum computed over an extended period of time can fail to reveal oscillatory activity that is temporally brief or that contains frequency non-stationarities.

The standard phase-amplitude coupling measure relying on Euler's formula failed to identify the pattern of cross-frequency coupling. The difficulty in this case was the source-level mixing in combination with weak gamma power relative to the noise spectrum. This simulation therefore also demonstrates how gedCFC can increase signal-to-noise characteristics, which is particularly important for non-invasilve measurements such as EEG.

As a proof-of-principle illustration, Method 2 was applied to human EEG data taken from a previously published study (*Cohen and van Gaal, 2013*), in which we reported that theta-alpha phase-amplitude coupling was modulated by error adaptation. In the present re-analysis, GED was applied to define a spatial filter that maximized theta-band power (peak 4 Hz, FWHM 4 Hz, based on the empirical peak theta frequency from channel FCz), which was then used to identify theta peaks and troughs. Next, the 64-channel data were filtered in the alpha band (peak 10.5 Hz, FWHM 4 Hz, based on the empirical alpha peak frequency from channel POz), and covariance matrices were computed using data surrounding ¼ of a theta cycle centered on peaks and troughs of the theta component (data were bandpass filtered here because of *a priori* hypotheses about theta-alpha coupling). The two components with the largest and smallest eigenvalues were selected as spatial filters. The power of those components was computed as a function of 30 theta phase bins, and the distributions are shown in *Figure 4c*. The relationship between alpha power and theta phase bin was shuffled 1,000 times to produce a null-hypothesis distribution. The 95% range of this distribution is shown as a gray patch in *Figure 4c*. Data values outside this range can be considered unlikely to occur by chance. Additional possibilities for statistical evaluation include fitting a sine wave to the alpha power distribution, or performing a test against a null hypothesis of a uniform distribution (e. g., a Kolmogorov-Smirnov test).

The main assumption underlying Method 2 is that the neural populations activated during peaks and troughs of the low-frequency rhythm produce field potential fluctuations that have separable projections onto multichannel recordings. Other assumptions are similar to those of Method 1: consistent spatiotemporal characteristics of each network, and a lower frequency dynamic that is rhythmic enough to be able to define peaks and troughs.

The main limitation of Method 2 is that the assumption of topographically separable networks according to different low-frequency phases is critical. If Method 2 were applied to a single network that varies only in amplitude according to lower-frequency phase, it would produce two identical (plus noise) covariance matrices. gedCFC would then return uninterpretable results (Method 1 would be appropriate in this case).

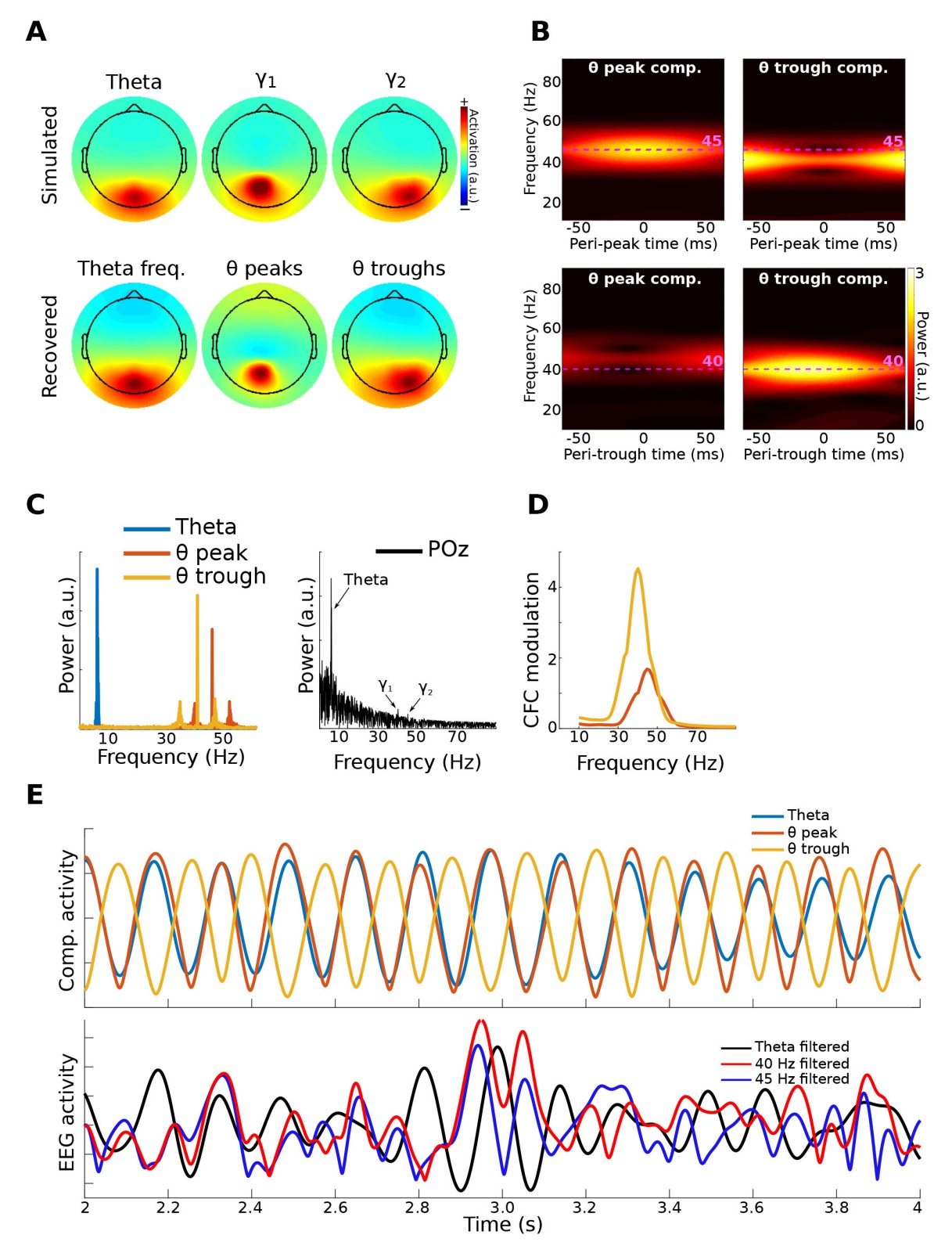

**Figure 3.** Method 2 applied to simulated data. Gamma power (40 and 45 Hz) in two brain dipoles was modulated by a theta wave from a third dipole. Activity from these three dipoles was projected, along with correlated 1/f noise from 2,001 other dipoles, to 64 scalp EEG channels (top row of (**A**) shows the signal-dipole projections). gedCFC was able to recover these three components ((**A**), bottom row) by comparing broadband covariance matrices between theta peaks and troughs. (**B**) shows the time-frequency power spectra of the components time series time-locked to theta peaks and

*Figure 3 continued on next page*

*Figure 3 continued*
theta troughs. (**C**) shows power spectra of the three component time series. For comparison, the power spectrum from POz (the channel with maximum power) is shown. (**D**) shows CFC modulation defined as the average peak-trough distances per frequency. (**E**) shows a 2 s snippet of data from the theta component and the power time courses from the theta-peak and theta-trough components. The lower panel shows data from POz bandpass filtered around theta, 40 Hz, and 45 Hz. Note that the continuous gamma power time series closely matched the theta wave from which they were simulated, although the covariance matrices were taken only from peaks and troughs.
The following figure supplement is available for figure 3:

**Figure supplement 1.** This figure shows a graphical overview of Method 2.

## Method 3: low-frequency waveform shape as a bias filter on sphered data

Methods 1 and 2 are robust to low-frequency non-sinusoidal oscillations precisely because the waveform shape is ignored, except to identify specific time points for time-locking the covariance matrices. On the other hand, very different waveform shapes could produce peaks at the same times. Given that waveform shape is the result of biophysical processes, one may wish to incorporate the lower-frequency waveform shape into the analysis. In this case, Methods 1 and 2 are inappropriate. Therefore, the goal of Method 3 is to use the lower-frequency time series as a continuous regressor without necessitating *a priori* specification of the relevant phase regions.

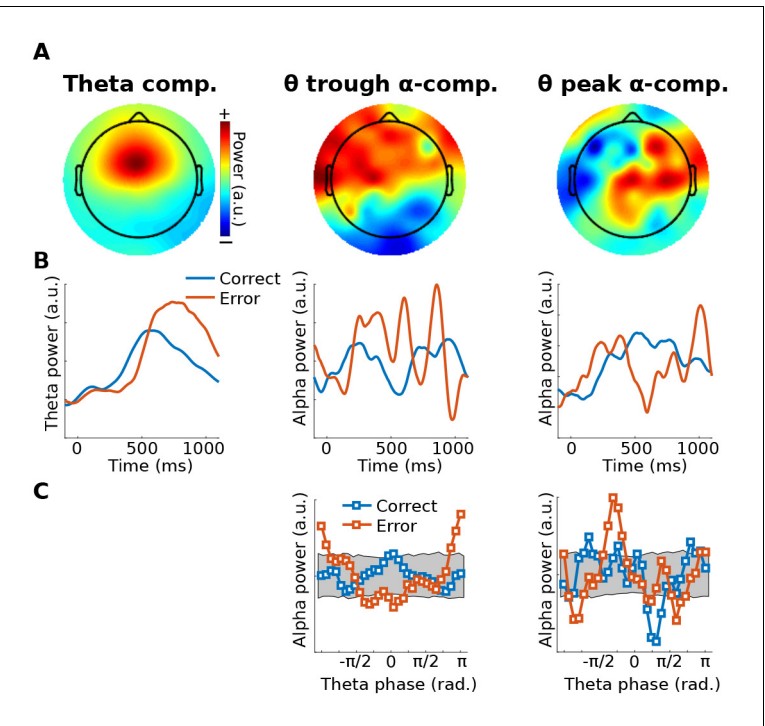

**Figure 4.** Illustration of Method 2 applied to real EEG data. The human volunteer (one subject selected from *Cohen and van Gaal (2013)*) performed a speeded reaction time task in which response errors frequently occurred. (**A**) shows topographical projections of the three components (theta power, theta trough, theta peak). (**B**) shows power time courses relative to stimulus onset (time = 0) from those components, separately for correct and error trials (power was extracted via the filter-Hilbert method). (**C**) shows how alpha power in the two alpha components fluctuated as a function of theta phase. Note that gedCFC was based on data from all trials pooled; the two experiment conditions were separated only when plotting. The gray bars in (**C**) show 95% confidence intervals based on 1,000 permutations in which the assignment between theta phase and alpha power was shuffled.

In Method 3, a low-frequency time series is used as a 'bias filter' that is applied to the multichannel data in order to determine the linear combination of electrodes that best matches the bias filter. The multichannel data are first sphered (i.e. whitened; after sphering, the covariance matrix is diagonal, meaning that inter-channel covariances are zero), which allows the filter to highlight only the spatiotemporal patterns that covary with the bias filter without influence from endogenous oscillatory activity.

The mechanics of Method 3 closely follow the 'joint decorrelation' method presented in *de Cheveigné and Parra (2014)*. A brief overview is provided here and depicted in *Figure 5—figure supplement 1*; interested readers are directed to the 2014 paper for more details. The first step is to extract the time series of a low-frequency component, as was done in Methods 1 and 2, to be used as a bias filter. The next step is to narrowband filter the multichannel data and extract the amplitude envelope (e.g., from filter-Hilbert or from Morlet wavelet convolution). This is an important step: the goal is to identify *power* fluctuations that vary according to the lower-frequency phase; therefore, it is the high-frequency power time series, not the bandpass filtered signal, that is of primary interest. The next step is to sphere the multichannel power fluctuations, which is implemented by scaling their eigenvalues. If the channels-by-time power time series data $X$ has eigenvectors and eigenvalues $V$ and $D$, then the sphered data are defined as $Y=X^T V D^{-\frac{1}{2}}$, where $^T$ indicates the matrix transpose and $^{-\frac{1}{2}}$ indicates the square root of the matrix inverse. Next, the bias filter is expanded into a Toeplitz matrix ($B$) that left-multiplies the data as $BY$. The eigendecomposition of the covariance matrix of $BY$ provides a new set of eigenvectors $W$, which are used to rotate the original eigenspace of the data as $V D^{-\frac{1}{2}} W$. The spatial filter with the largest eigenvalue is applied to the non-sphered narrow-band multichannel power time series data to obtain the component. This procedure is repeated over a range of higher frequencies, using the same $B$ matrix for all frequencies.

Data for Method 3 were simulated like those for Method 1. *Figure 5* shows the fit of the filtered component to the simulated dipole gamma power time series ($R^2$). Fits were equivalently good (up to 5) when using correlations and phase synchronization measures, and were poor (<0.1) when using data from the gamma-maximum electrode instead of the component. The theta peak-locked power spectra from the component showed a precise spectral-temporal localization consistent with how the data were simulated.

Empirical data for a proof-of-principle demonstration were taken from recordings in a human epilepsy patient. Eleven channels were simultaneously recorded, ten along the medial temporal lobe axis (including the amygdala and hippocampus, locally-average reference) and one from surface channel Cz (on the vertex on the scalp, referenced to mastoids). Low-frequency midline cortical activity (peak frequency 4 Hz, FWHM 3 Hz) was extracted from Cz and was used as the bias filter in the Toeplitz matrix $B$. To reduce computational load and increase signal-to-noise characteristics, the continuous data were cut into 50 equally sized epochs of 4 s each, and the covariance matrix of $BY$ was computed separately per epoch and then averaged together.

The gedCFC was applied iteratively using power time series from the medial temporal lobe in 70 frequencies, ranging from 10 to 120 Hz. CFC was quantified using $R^2$ fit between the component power time series and the Cz low-frequency rhythm, and as phase synchronization (which ignores the amplitude fluctuations). *Figure 6a* reveals a peak in this coupling at around 65 Hz. Inspection of time-frequency power spectra time-locked to theta peaks (*Figure 6b*) indicates that high-frequency power built up prior to the Cz theta peak. This component was more strongly driven by anterior channels (*Figure 6c*). Statistical evaluation of the coupling-by-frequency analysis was implemented by shifting the low-frequency time series by a random amount relative to the power time series, and by recomputing the correlation and phase synchronization coupling measures. This procedure was repeated 1,000 times to generate a null hypothesis distribution. The 99% values of these distributions are shown as dotted lines in *Figure 6a*.

Interactions between the medial temporal lobe (and hippocampus in particular) and prefrontal cortex are widely implicated in memory formation (*Preston and Eichenbaum, 2013*). A multivariate technique like gedCFC might be useful in providing insights into how different hippocampal-prefrontal networks are involved in different aspects of memory (*Shin and Jadhav, 2016*). Incorporating the waveform shape into the filter might also prove insightful considering that rat hippocampal theta is non-sinusoidal (*Belluscio et al., 2012*).

The key assumption of Method 3 is that the lower-frequency waveform shape is important (as opposed to being used simply to identify peaks and troughs). The main limitation is that this method

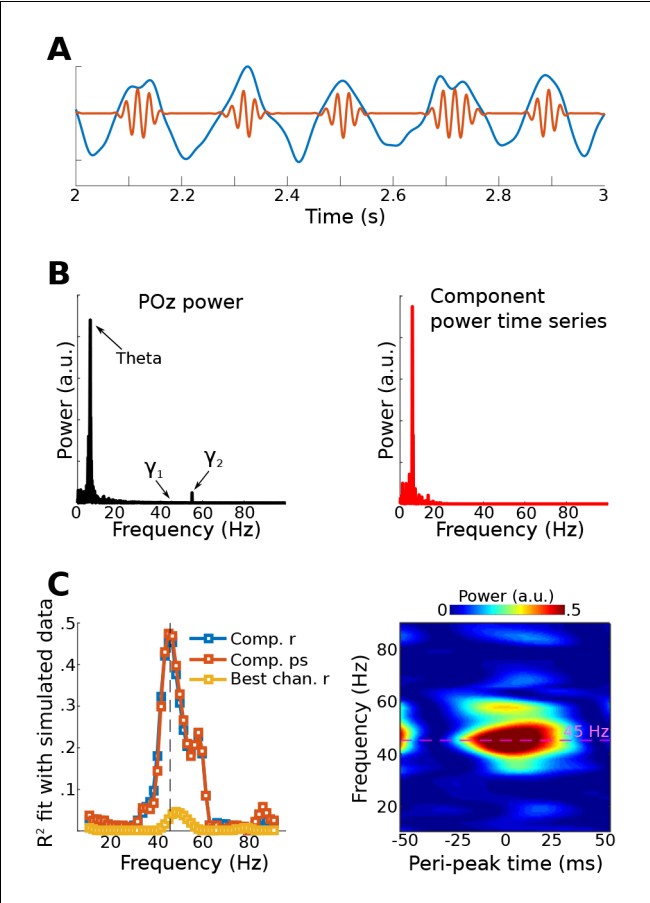

**Figure 5.** Results from simulation for Method 3. Data were simulated in the same dipoles used in Method 1. (**A**) shows an example of the dipole time series, illustrating the theta wave and the gamma modulation (scaled for visibility). Power time series were extracted from all channels, and the low-frequency time series was used as a bias filter against the multichannel power time series matrix. (**B**) shows the power spectrum from channel POz (left panel). Note the absence of a prominent 45 Hz peak. The right panel shows the power spectrum of the 45 Hz power time series from the gedCFC component. The peak at 6 Hz reflects the modulation of 45 Hz power by the theta rhythm. (**C**) shows the fit (correlation or phase synchronization) between the gedCFC component per frequency and the simulated power time series. For comparison, the correlation result was also performed at channel POz. The right panel shows the time-frequency power spectrum locked to theta peaks.

The following figure supplement is available for figure 5:

**Figure supplement 1.** Graphical overview of the procedure for applying joint decorrelation adapted to Method 3.

relies crucially on proper specification of the bias filter. The bias filter could be misspecified if its spatial and spectral characteristics are not known or are suboptimally estimated. Thus, Method 3 should be used only when there is a strong *a priori* motivation to justify using an appropriate bias filter.

## Method 4: gedCFC using time-delay embedded matrices

In Methods 1–3, the higher-frequency components are computed using covariance matrices, but their temporal and spectral characteristics are computed using sinusoid-based filters, which impose sinusoidality and temporal symmetry on the results, particularly when using narrowband filters. Therefore, the purpose of Method 4 is to empirically compute the higher-frequency waveform shape, as well as its spatial distribution, directly from the data, without relying on narrowband filters. This allows the identification of CFC-related activity that is not necessarily sinusoidal or even

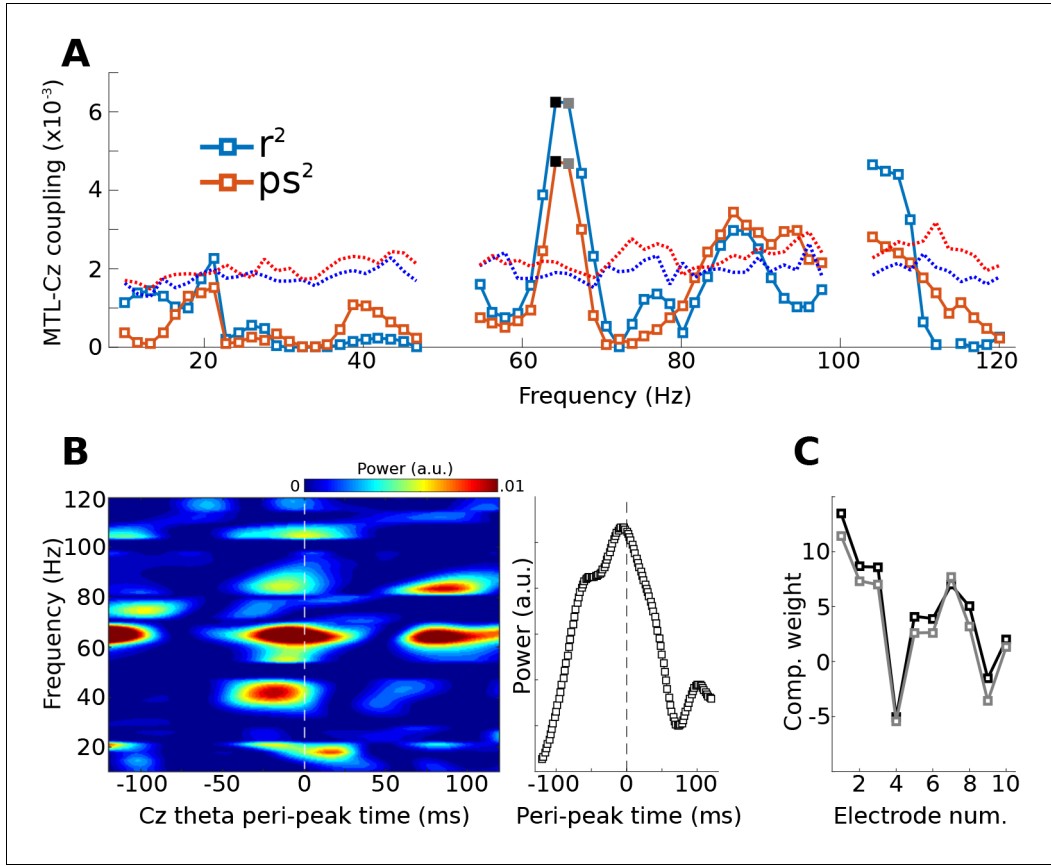

**Figure 6.** Method 3 applied to empirical data from a human epilepsy patient, with ten channels in the medial temporal lobe (MTL) and Cz, a scalp electrode that measures midline frontal activity. Covariance matrices of MTL channels were created locked to peaks from Cz filtered at 4 Hz. (**A**) shows the coupling strengths, measured both as correlations (r) and as phase synchronization (ps) over a range of frequencies (notch-filtered data at 50 Hz and 100 Hz are omitted). Correlation values were squared to avoid sign issues, and phase synchronization values were squared for comparability. The dotted lines indicate the 99% confidence intervals based on 1,000 permuted shufflings. (**B**) shows the time-frequency power spectrum of the MTL component time-locked to Cz theta peaks. The right plot shows the time course of activity averaged around 65 Hz (frequencies selected based on visual inspection of (**A**). (**C**) illustrates the component weighting across the ten MTL electrodes from two neighboring frequencies (see black and gray filled squares in (**A**); smaller numbers are anterior).

rhythmic. And if the activity is sinusoidal, the spatiotemporal filter will empirically identify the sinusoidality without the confound of using a sinusoidal filter.

Method 4 is accomplished through time-delay embedding, which means that additional rows are added to the data matrix, which correspond to time-shifted versions of the channel time series. For example, a first-order time-delay embedded matrix of 64 electrodes would have 128 rows; the first 64 rows are taken from the 64 electrodes and time points 1 to end-1, while the next 64 rows are taken from the same 64 electrodes and time points two to end. Importantly, because GED doesn't 'know' that the data come from time-delayed versions of the same electrodes, the weights assigned to time-delayed rows act as temporal weights from which a temporal filter is empirically computed (*de Cheveigné, 2010*). For example, a filter that finds the temporal derivative would have weights of 1 and −1 for the original and time-delayed rows. In practice, more than one delay embedding is useful. Time-delay embedding was developed in dynamical systems analysis, and has been successfully applied to psychometric and neural data (*Brunton et al., 2016*; *de Cheveigné and Simon, 2007*; *Lainscsek and Sejnowski, 2015*; *Tome et al., 2004*; *von Oertzen and Boker, 2010*). After creating this time-delay embedded data matrix, Method 4 proceeds similarly as Method 1: two covariance matrices are computed, one from data surrounding troughs and one from the entire time series, and

GED is applied to those two matrices. The eigenvector with the largest eigenvalue is a spatiotemporal filter of size $1 \times MN$, where M is the number of channels and N is the number of embeddings. To interpret this filter, its forward model can be reshaped to $M \times N$ and visualized as a time-by-channels matrix.

The number and spacing of delay embeds are parameters of the time-delay-embedded matrix. If one wishes to use the Fourier transform to determine the filter's frequency characteristics, the delays should be linear (that is, embedding 1, 2, 3, 4, instead of 1, 2, 4, 8), and the number of delay embeds determines the frequency resolution. Here, a 60th order matrix was used, meaning the data matrix contains 3,840 rows (60 embeddings times 64 channels). At 1,024 Hz, this provided a frequency resolution of 17 Hz, which was then zero-padded to a frequency resolution of 2 Hz. There are other methods to determine the embedding dimension (*Cao, 1997*; *Maus and Sprott, 2011*); it is beyond the scope of this paper to include an exhaustive discussion of these methods.

Data were simulated by adding a burst of 75 Hz activity to one dipole at each trough of theta from another dipole (*Figure 7a*), then adding 1/f noise and projecting to 64 EEG electrodes. The gamma power bursts were time-locked to the troughs, but the phase of the gamma burst varied randomly across bursts, thus producing a non-phase-locked (i.e., 'induced') response. A high-pass temporal filter (lower edge 20 Hz) was applied to the data to prevent the component from simply reflecting the shape of the lower-frequency time series around the peak.

*Figure 7* shows that the theta and gamma components were accurately recovered. This result can be compared with the GED results to the trough-triggered averages (this is an event-related potential, where the 'event' is the trough), which failed to detect the simulated cross-frequency coupling. This occurred even without adding noise to the data, and is attributable to the non-phase-locked nature of the gamma bursts. Note that, as with previous simulations, the gamma bursts were generally too small and transient to be visually detected in the channel power spectrum, even at the electrode with the maximal theta-trough-component projection (*Figure 7D*).

Method 4 was applied to empirical data taken from the rat hippocampus (data downloaded from crcns.org; *Diba and Buzsáki, 2008*; *Mizuseki et al., 2013*; dataset ec013.156). A theta component was obtained by comparing covariance matrices between 8 Hz and the average of 4 Hz and 12 Hz (a broadband reference was not used because it was dominated by theta). As with the simulation, the data were high-pass filtered at 20 Hz to prevent the largest components from reflecting theta phase. Peri-peak covariance matrices were computed, and compared against the total covariance. The eigenvector with the largest eigenvalue identified a component with a spectral peak in the gamma band that had a similar spatial projection onto all channels (*Figure 8*). By contrast, the peak-locked average had a complex waveform shape without a clear spectral concentration.

As implemented here, Method 4 assumes a single network that varies according to low-frequency phase. The method could be adapted to identify two networks, as with Method 2. Parameterizing the matrix is also important, as insufficient delay embeds will reduce the temporal precision and frequency resolution.

The primary limitation is that working with delay-embedded matrices requires sufficient computing resources. For example, two minutes of 256 channels at 1 kHz and a 40-fold embedding would produce a matrix of size 10,240-by-120,000. Matrix operations can be slow and can produce numerical inaccuracies. Two potential solutions are to temporally downsample the data or to use a data-reduction technique, such as principal components analysis, to reduce the data, to 40 dimensions for example., makng it possible to perform analyses in the 40-dimensional subspace.

## Method 5: gedCFC for spike-field coherence

Spike-field coherence refers to local field potential (LFP) correlates of single-cell spiking. It is often quantified either through spike-triggered averages, in which the LFP traces are averaged around each spike (similar to an event-related potential, where the event is the spike), or through phase-clustering-based methods, in which the phase angles are extracted from frequency-specific analytic time series (e.g., from complex wavelet convolution).

Although spike-field coherence is not typically conceptualized as a manifestation of CFC, in the gedCFC framework, spike-field coherence is simply a special case in which multichannel data are time-locked to the action potential instead of to a low-frequency trough. In this sense, any of the methods presented above (or combinations thereof) can be adapted to work for multichannel spike-field coherence. In this section, an adaptation of Method 4 is demonstrated. Method 4 is highlighted

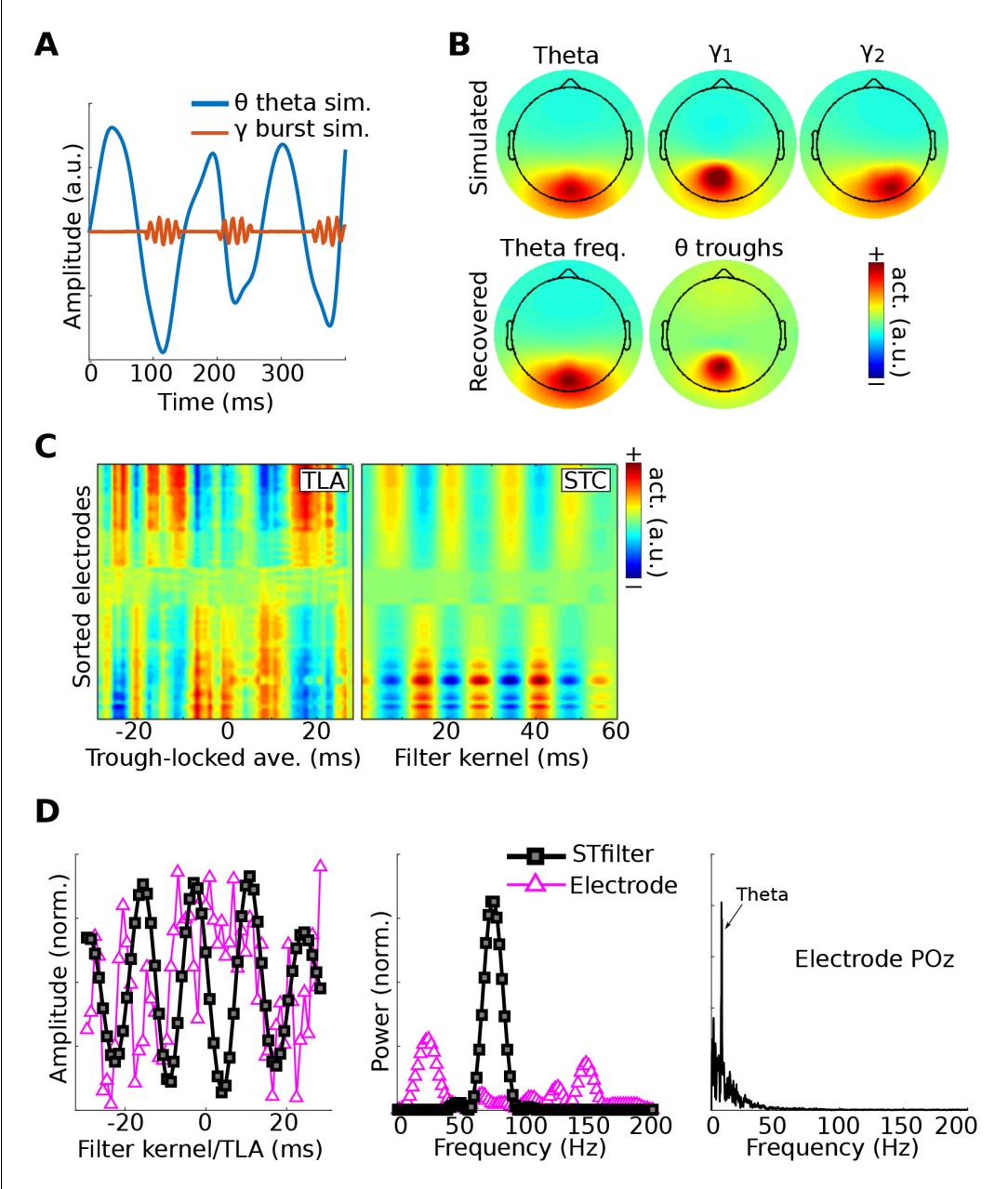

**Figure 7.** Method 4 applied to simulated EEG data. Bursts of 75 Hz gamma in one dipole were locked to theta troughs in a different dipole (**A**). Activity in these two dipoles, along with a 'distractor' gamma signal at 50 Hz and correlated 1/f noise at all other dipoles, was projected to scalp EEG channels. (**B**) shows the dipole projections and their reconstructions from gedCFC on delay-embedded matrices. (**C**) shows the trough-locked average (TLA) over time for all electrodes (electrodes are sorted according to spatial location, with anterior electrodes on top and posterior electrodes on the bottom) on the left and the spatiotemporal component (STC) extracted via gedCFC on the right. (**D**) shows the time series of the filter kernel from electrode POz (left), the filter kernel power spectrum (middle), and the power spectrum of POz activity (right; note the absence of a pronounced peak at 75 Hz). Even with only minimal noise, the non-phase-locked nature of the gamma burst prevented the TLA from revealing any meaningful relationship (phase-locking does not affect the single-trial covariance matrices). Amplitudes are normalized to facilitate direct comparisons.

here because it allows unconstrained empirical discovery of asymmetric spike-related LFP activity; it can be used, for example, if the LFP waveform shape differs before and after the spike.

Simulated data were modeled after a 16-channel linear probe that is often used to study laminar dynamics in the cortex. Spikes were generated at random times, and bursts of 70 Hz gamma were

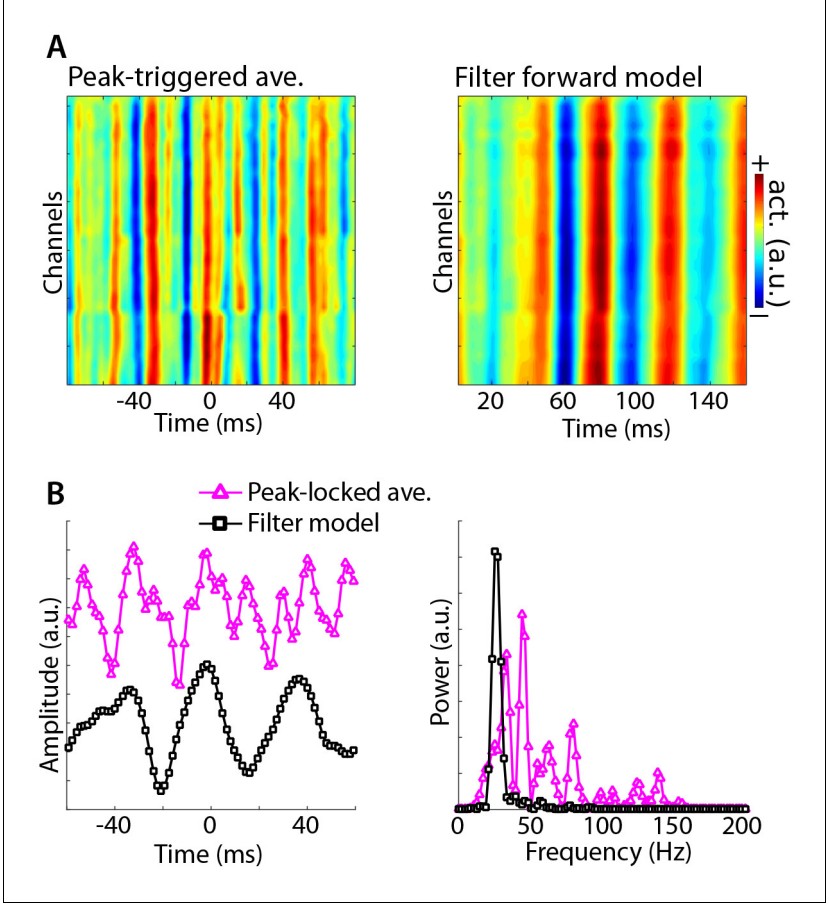

**Figure 8.** Illustration of Method 4 applied to empirical data from rodent hippocampus. GED was used to identify a theta component (peak frequency 8 Hz), and peak times were identified. (**A**) illustrates the peak-triggered LFP trace over 32 channels (high-pass filtered at 20 Hz), and the forward model of the first gedCFC component. (**B**) illustrates the peak-locked average and the component model, averaged over all channels, in the time domain (left panel) and in the frequency domain (right panel). Amplitudes were scaled and y-axis-shifted for comparability. DOI: 10.7554/eLife.21792.015

centered on each spike, with a 'rugged' laminar profile (similar to that in *Figure 1—figure supplement 3*, which illustrates that gedCFC makes no assumptions about spatial smoothness) (*Figure 9a*). In addition, a 75 Hz gamma rhythm that was uncorrelated with spike timing was added as a 'distractor.' Random 1/f noise was also added.

As expected, Method 5 recovered the simulated spatiotemporal LFP pattern (*Figure 9c–d*). With few spikes (N = 40), only a small amount of noise will prevent the spike-triggered average from revealing the true effect. Even when no noise was added, the spike-triggered average failed to capture the true effect if the gamma burst was time-locked but not phase-locked to the spike.

Method 5 was applied to an empirical dataset from recordings of rat medial prefrontal cortex and hippocampus (data downloaded from CRCNS.org; *Fujisawa et al., 2008*, *2015*; dataset EE.042). Data were recorded from 64 channels in the medial prefrontal cortex and 32 channels in the hippocampus. Spiking data were taken from a single neuron in the prefrontal cortex. The first several components appeared to capture a spike artifact in the LFP (*Figure 10a*). Two later components were selected on the basis of visual inspection. Neither component appeared to suffer from spike artifacts, and both revealed rhythmic LFP dynamics surrounding the spike with phase offsets between the prefrontal and hippocampal electrodes. Temporal and spectral plots of the filter forward model, as well as the spike-triggered average component time series (*Figure 10b*), revealed that these two spike-field networks had different characteristic frequencies and temporal dynamics. Note the temporal asymmetries (before vs. after the spike) and nonstationarities in the component

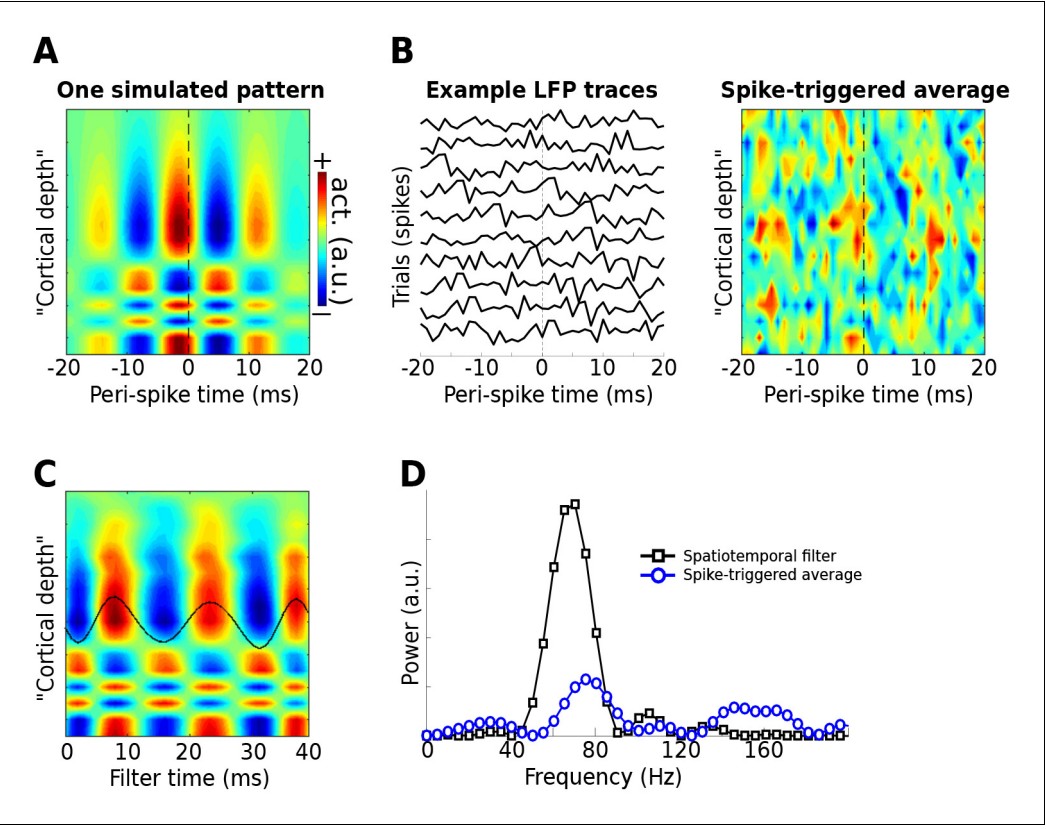

**Figure 9.** Use of the gedCFC framework for detecting spike-field coherence in multichannel data. LFPs were simulated in 16 channels, modeled after a silicon probe often used to measure cortical laminar activity. Forty random spike times were generated, and complex spatiotemporal patterns were time-locked to each spike time. (A) illustrates the basic pattern, which was phase-randomized on each spike. (B) illustrates a few single-spike LFP traces from one channel, and the spike-triggered average over time and space (compare with the template in (A)). (C) shows the forward model from the largest gedCFC component reshaped to a 2D matrix. (D) shows the power spectrum of the component and the spike-triggered average (power spectra averaged across channels).

time series; these are readily visible because no filters were applied that would artificially impose sinusoidality or acausality on the data; instead, the filter kernel was empirically estimated on the basis of broadband data.

Components that capture spike artifacts can be identified by visual inspection or by examining the power spectrum of the filter forward model: artifacts have higher energy in a broad high-frequency range relative to lower frequencies, whereas true effects should have higher energy below ~200 Hz compared to above. Non-artifact components can be individually selected on the basis of prespecified criteria such as frequency band, or can be summarized using principal components analysis.

*Figure 10* shows one example of how to apply the gedCFC framework to investigate multichannel spike-field coherence. There are several other possibilities. One could identify the spike artifact components and reconstruct the data with the artifact removed (*de Cheveigné, 2010*) in order to apply traditional spike-field coherence analyses. Another possibility is to adapt Method 2 to identify LFP components corresponding to different temporal patterns of spiking, such as singlets vs. trains of action potentials. An example of this approach is presented in *Figure 10—figure supplement 1*.

## Over-fitting and null-hypothesis testing

Components analyses that are guided by minimization or maximization criteria (such as GED) entail a risk of overfitting. Essentially, one is searching through a high-dimensional space for a particular

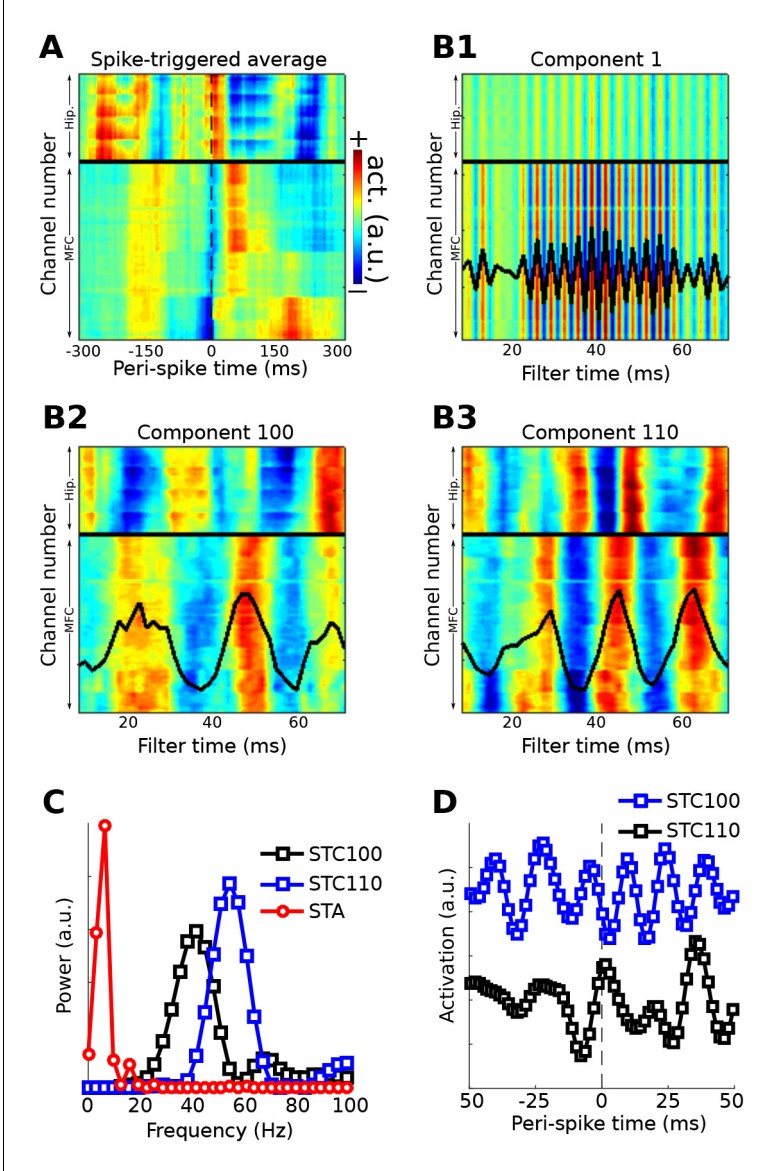

**Figure 10.** Method 5 applied to rodent hippocampal-prefrontal single-cell and LFP recordings. (**A**) shows the spike-triggered average across all channels (MFC = medial frontal cortex; Hip. = hippocampus). (**B**) illustrates the forward models of several gedCFC components. The first few components captured the spike artifact, as shown in (**B1**). Later components reflected different aspects of physiological activity, two of which are illustrated here ((**B2**) and (**B3**)). The power spectra of the spike-triggered average and the two physiological components are shown in (**C**) (the component spectra have the same y-axis scale; the spike-triggered average was scaled down for comparability; STC = spatiotemporal component). (**D**) shows the spike-triggered average of the two components (shifted on the y-axis for comparability). Note that the components are temporally asymmetric and nonsinusoidal; their waveform shapes are defined empirically without imposition of a sinusoidal filter.

The following figure supplement is available for figure 10:

**Figure supplement 1.** This figure shows another possible application of the gedCFC framework for multivariate spike-field coherence, using the same dataset as used in *Figure 10*.

feature; even in pure noise, the components will identify some pattern that best fits the criteria. There are at least three appropriate statistical methods for evaluating the results of gedCFC.

One approach is to compare the resulting components against a null-hypothesis distribution. This null-hypothesis distribution can be derived by applying the method but using randomly selecting time points to be 'troughs' (the total number of randomized 'troughs' should be the same as the number of real troughs). When repeated, for example 1,000 times with different sets of random 'troughs,' the spatiotemporal characteristics, including time courses and power spectra, of the true components can be compared against the empirical null hypothesis distribution. This procedure follows the non-parametric permutation framework that is widely used in neuroimaging and electrophysiology (*Cohen, 2014*; *Maris and Oostenveld, 2007*). A difficulty with this approach is that the resulting components may capture the low-frequency component. One could apply a high-pass filter or ignore any coupling close to the frequency used to extract the low-frequency component.

A second approach is to apply gedCFC to data pooled across all experiment conditions, and then to apply statistical comparisons of component time series or power spectra across different conditions. In this case, although the component-fitting procedure itself may include overfitting, differences across conditions are not biased because the filter was defined orthogonal to condition differences. A related approach is to apply the method to a range of frequencies and then to test whether the coupling strength at different frequencies is statistically significant, as illustrated in *Figure 6*.

A third approach is to compute the gedCFC filter from independent data and then apply the filter to the experimental data. Independent data could be drawn from different trials from the main experiment (as in cross-validation) or from an independent dataset such as a localizer. In the case of cross-validation, confidence intervals can be computed.

## Discussion

There is spatiotemporal structure in the mesoscale electrical activity of the brain, which reflects the dynamic interactions within and across populations of neurons. Because this spatiotemporally structured activity is larger than that which can be recorded by a single electrode, single-electrode analyses will miss or underestimate these patterns. Therefore, as the number of simultaneous electrodes used in neural recordings increases (*Stevenson and Kording, 2011*), so does the need to consider methods for extracting the neural patterns that underlie those recordings. Component-extraction and dimensionality-reduction methods have proven useful in neuroscience, both in terms of managing large-scale data sets and in terms of providing insights into neural mechanics that are difficult to obtain only from visualizing topographical maps (*Cunningham and Yu, 2014*; *Onton et al., 2006*).

### Cross-frequency coupling in neural time series data

CFC is a fascinating observation that has garnered interest amongst empiricists, modelers, theoreticians, and mathematicians. It has also been the subject of considerable debate, as there seems to be a growing number of publications criticizing CFC methods (*Aru et al., 2015*; *Hyafil, 2015*; *Lozano-Soldevilla et al., 2016*). Many of these methodological concerns affect only one framework for analyzing CFC using individual electrodes via narrow bandpass filtering and applying Euler's formula, phase synchronization, or other distribution analyses. These methods provide accurate results under the assumption of uniform phase angle distribution; CFC biases result from violations of this assumption (*van Driel et al., 2015*). The framework introduced here evades several of these problems. This new framework does not invalidate or antiquate existing CFC measures; traditional CFC methods are well-suited for situations in which the lower and higher frequency dynamics are sinusoidal and are produced by neural circuits that have the same spatial projections to multichannel recordings.

The advantages of gedCFC go beyond evading potential biases in traditional CFC measures. This multivariate framework allows detection of diverse spatiotemporal manifestations of CFC, some of which may be difficult to detect with other methods. The inspiration for this framework came from considering the physics of M/EEG and LFP field propagation, by theoretical predictions that the low-frequency rhythm should regulate anatomically diverse neural populations, and by developments in spatial filtering methods used in source separation and dimensionality reduction analyses.

## Advantages of gedCFC

Several specific advantages were highlighted in the Results subsections; a few general remarks are made here. One advantage of the gedCFC framework is that there are no assumptions about underlying generative models that may have produced the data (other than the plausible assumption that electromagnetic fields propagate simultaneously and linearly to multiple channels). It is not necessary to assume Gaussian or Poisson processes for example, nor is it necessary to employ complicated statistical or biophysical models. Instead, the components are extracted directly from the empirical data.

A second advantage is flexibility. From the single equation **SW=RWΛ**, five applications were derived. By carefully selecting parameters, frequency ranges, experiment conditions, time windows, and so on, this framework is easily extended and tailored to specific hypotheses. One should be cognizant that components-based and dimensionality-reduction-based analyses will perform well when there are clean data and when the researcher has clear and physiologically motivated objectives (*van Ede and Maris, 2016*). Excessively noisy data or poorly specified analysis goals (i.e., from poorly specific hypotheses) may produce misleading or uninterpretable results.

A third advantage is increased signal-to-noise characteristics (and therefore, increased sensitivity to detecting CFC), which come from analyzing a weighted combination of data from all electrodes instead of a single electrode. The key is to create those weights appropriately and in an hypothesis-driven manner. The presence of a visually robust peak in the FFT-derived power spectrum is not a hard constraint on the success of gedCFC (as illustrated in *Figures 3* and *5*). An oscillation may exist in a data set and yet have a small peak in a static power spectrum due to non-stationarities in power and phase characteristics.

A fourth advantage is computation time. Many traditional CFC analyses can be prohibitively time-consuming. gedCFC is fast because software programs like MATLAB use efficient libraries for estimating eigenvalues. On most computers and for most covariance matrices, the MATLAB eig function takes a few milliseconds. The exception is Method 4, which can be processor-intensive for very large matrices (the Results section provided a few suggestions for improving computation time, including temporal downsampling and dimensionality reduction). Bandpass filtering, when used, is also fast, and the narrowband filter used here involves only one FFT and one IFFT.

## Limitations of gedCFC

Specific limitations were discussed in each Results subsection; a few additional remarks are made here. First, sign-flipping of components produces uncertainty of peak vs. trough in the low-frequency component. The signing strategy used here is discussed in the Materials and methods section. Relatedly, bursts of high-frequency activity might be locked to the rising slope or falling slope of the low-frequency rhythm instead of the peak or trough (*Fujisawa and Buzsáki, 2011*). Careful data examination and *a priori* theoretical guidance should be used to determine the appropriate phase values for time-locking.

A second limitation is that the component with the largest eigenvalue is not necessarily the only theoretically relevant component (as illustrated in *Figure 10*, the largest eigenvalued component may contain artifacts). It is possible that the first few components define a subspace of physiological CFC dynamics. In other words, a single component does not necessarily correspond to a single brain dynamic (*de Cheveigné and Parra, 2014*). It is advisable to inspect several components for physiological interpretability and modulation by experimental conditions, as was done in *Figures 2* and *10*.

Third, the gedCFC framework is designed for two covariance matrices. To identify more than two networks, one could iteratively apply, for example, Method 1 to six different phase bins. The six spatial filters could be applied the data to reconstruct six different component time courses. This approach is valid if there really are six distinct networks; otherwise, the same spatial filter might be recreated multiple times, with apparent differences attributable to over-fitting noise.

Finally, gedCFC should not be treated as a 'black-box' analysis procedure in which results are interpreted without careful inspection of the data, analysis procedures, and parameters. Misleading or uninterpretable results can occur if excessively noisy data or inappropriate parameters are applied. Instead, gedCFC is best conceptualized as an analysis strategy that mixes physiologically inspired hypothesis testing (e.g., when defining the lower-frequency dynamic) with

blind network discovery. The discovery aspect stems from not needing to specify which topographical regions or frequency bands will manifest CFC.

## Nonstationarities and condition differences

The intrinsic nonstationarities of the brain can be problematic for analyses that rely on stationarity over long periods of time, and are also problematic for traditional CFC analyses. Sinusoidal stationarity is not assumed in the gedCFC framework. gedCFC is also not affected by phase lags across different electrodes, as long as the lags are consistent. The core assumption is that the neural patterns maintain a consistent spatiotemporal signature, for example, that each trough-locked covariance matrix is a representative sample. If narrowband filtering is applied to the higher-frequency activity, then an additional assumption is that the activity can be reasonably approximated by sinusoidal basis functions (this is generally a reasonable assumption at the scale of hundreds of ms).

Task-related experiments often have multiple conditions. This leads to the decision to compute covariance matrices separately for each condition, or from condition-pooled data to which the spatial filter is then applied to each condition separately. If the covariance structure is expected to be qualitatively different across conditions, separate covariance matrices should be preferred. On the other hand, if the network structure is expected to be the same and only the strength of the modulations are expected to vary, it is preferable to pool data across all conditions when computing the GED.

The quality of gedCFC components is related to the quality of the covariance matrices, and a sufficient amount of clean data ensures high-quality covariance matrices. Determining the quality of a covariance matrix can be difficult. One metric is the condition number, which is the ratio between the largest and smallest singular values of the matrix. However, there is no hard threshold for considering a matrix to be ill-conditioned, and GED can be successfully applied to singular matrices (which have a condition number of infinity). When working with a small amount of data, using alternative methods of estimating covariance, such as shrinkage estimators, may be helpful (*Daniels and Kass, 2001*).

## gedCFC for spike-field coherence provides new insights and avoids potential artifacts

One of the main confounds in spike-field coherence analyses is that the brief large-amplitude spike can cause an artifact in the low-pass filtered LFP (*Ray, 2015*; *Zanos et al., 2011*). This is a well-known phenomenon in signal processing that is often called an edge artifact.

Spike-field coherence here was investigated by adapting Method 4 (*Figures 9* and *10*) to give rise to Method 5 (*Figure 10—figure supplement 1*), but other situations might call for different approaches. For example, Method 2 could be further adapted to identify multivariate components that differentiate spike-field coherence patterns between two cognitive or behavioral states (e.g., attend vs. ignore, or rest vs. run). A primary advantage of using gedCFC for spike-field coherence is that traditional spike-field coherence analyses can detect only phase-locked activity; gedCFC will additionally detect non-phase-locked responses, such as a spike-locked burst of non-phase-locked gamma.

## Further advances

GED is a powerful technique, but it is not the only source-separation method. Many other components-based and dimensionality-reduction-based methods exist, including iterative and nonlinear algorithms (*Jutten and Karhunen, 2004*). For example, phase-amplitude coupling can be identified using parallel factor analysis or tensor decomposition (*van der Meij et al., 2012*). A primary difference between gedCFC and such techniques is that gedCFC is a hypothesis-driven approach, whereas other techniques are blind decomposition methods that are well-suited for exploratory analyses. Another important advantage of gedCFC is that it is straight-forward both conceptually and in implementation, and therefore it will facilitate analyses for researchers with diverse backgrounds and expertise in mathematics and programming. Furthermore, linear methods tend to produce robust and easily interpretable results (*Parra et al., 2005*). Nonetheless, the goal of this paper was not to argue that gedCFC is the only source-separation algorithm applicable to CFC. Instead, the goal was to highlight the significant insights into the neural mechanisms and the implications of CFC that can

be gained by expanding the repertoire of data analyses, as well as theoretical conceptualizations, from a single-electrode-sinusoid-based framework to a multivariate-components-based framework.

## Materials and methods

### Simulated EEG data

Simulated EEG data were created by generating time series in 2,004 dipoles in the brain according to different assumptions of CFC dynamics as described in the Results, and then projecting those activities to virtual scalp electrodes. Random data were generated by taking the inverse Fourier transform of random complex numbers sampled from a uniform distribution. A 1/f shape was imposed by tapering the spectrum by a sigmoidal curve and then concatenating a mirrored version of the tapered spectrum to produce the negative frequencies. This procedure was done separately for each voxel, thereby producing 2,004 uncorrelated dipoles.

Next, cross-voxel correlations were imposed across all dipoles by creating a random dipole-to-dipole correlation matrix with a maximum correlation of .8, and computing the new data as $\mathbf{Y} = \mathbf{X}^T\mathbf{VD}^{\frac{1}{2}}$, where $\mathbf{V}$ and $\mathbf{D}$ are matrices of eigenvectors and eigenvalues of the correlation matrix, $\frac{1}{2}$ indicates the square root, and $\mathbf{X}$ is the data matrix. Finally, time series data from selected dipoles were replaced with sinusoid-like time series in the theta or gamma bands, as described in the Results section.

Dipole locations were based on a standard MRI brain. The forward model to project three cardinal directions at each dipole location to the scalp EEG channels was created using algorithms developed by openmeeg (*Gramfort et al., 2010*) and implemented in the Brainstorm toolbox in MATLAB (*Tadel et al., 2011*). MATLAB code to generate the simulations, and to apply all five methods, can be found online (mikexcohen.com/data).

### Empirical datasets

The following text is copied verbatim from the Human Connectome Project, as requested: MEG data in *Figure 2* 'were provided by the Human Connectome Project, WU-Minn Consortium (Principal Investigators: David Van Essen and Kamil Ugurbil; 1U54MH091657) funded by the 16 NIH Institutes and Centers that support the NIH Blueprint for Neuroscience Research; and by the McDonnell Center for Systems Neuroscience at Washington University.' The dataset used here was resting-state data from subject 104012_MEG, session 3.

EEG data in *Figure 4* were from a single subject randomly selected from Cohen and van Gaal (*Cohen and van Gaal, 2013*). The human intracranial EEG data in *Figure 6* were taken from 200 s of resting-state data recorded in a patient with epilepsy, who had electrodes implanted as part of pre-surgical mapping. Data were acquired from the Department of Epileptology at the University Hospital of Bonn, and informed consent was acquired for the recording.

Rodent LFP and single-unit recording data were downloaded from crcns.org. The appropriate references and dataset identifiers are cited in the Results section.

### Visualizing gedCFC components

The topographical projections of the components were obtained from columns of the inverse of the transpose of the eigenvectors matrix (*Haufe et al., 2014*). For the MEG data, projections to the brain were computed using a procedure described in *Hild and Nagarajan (2007)*, and adapted by *Cohen and Gulbinaite, 2016*. The forward model for this subject was computed using the Brainstorm toolbox. Brain voxels with values lower than the median value across all 15,002 voxels were not colored.

### Sign-flipping of components

One issue that arises in eigenvector-based components analysis is that the sign of a vector is often not meaningful — the eigenvector points along a dimension, and stretching, compressing, or flipping the sign of the vector does not change the dimension. When extracting power, the sign of the time-domain signal is irrelevant. For topographical projections, the sign also doesn't matter, although for visual clarity, the sign was adjusted so that the electrode with the largest magnitude was forced to be positive (this is a common procedure in principal components analysis).

However, identifying 'peaks' vs. 'troughs' clearly requires the correct sign. The solution used here was to correlate the low-frequency GED-based component with the bandpass filtered data at the electrode with the maximal topographical projection. If the correlation coefficient had a negative sign, the sign of the low-frequency component was flipped. This solution is not guaranteed to produce the correct answer, particularly with noisy data. If the interpretation of 'peak' vs. 'trough' is crucial, it is advisable to run gedCFC separately for peaks and for troughs.

### Narrowband temporal filtering

Temporal filtering was done by circular convolution. The Fourier transform of the data was computed using the MATLAB function fft, and was pointwise multiplied by a frequency-domain Gaussian, defined as $exp(-0.5[x/s]^2)$, where $exp$ indicates the natural exponential, $x$ is a vector of frequencies shifted such that the desired peak frequency of the filter corresponds to 0, and $s$ is $\sigma(2\pi - 1)/(4\pi)$, where $\sigma$ is the full width at half maximum of the filter, specified in Hz. This filter is advantageous over equivalently narrow FIR or IIR filters because it contains fewer parameters and has no sharp edges in the frequency domain, and therefore does not require careful inspection of the filter kernel's temporal and spectral response profiles. The inverse Fourier transform was then applied to the tapered frequency domain signal to get back to the time domain. When power time series were required, the Hilbert transform was applied and its magnitude was taken. MATLAB code to implement this filter is provided with the online code.

## Additional information

### Funding

| Funder | Grant reference number | Author |
|---|---|---|
| European Research Council | ERC-StG 638589 | Michael X Cohen |

The funders had no role in study design, data collection and interpretation, or the decision to submit the work for publication.

### Author contributions

MXC, Conception and design, Acquisition of data, Analysis and interpretation of data, Drafting or revising the article, Contributed unpublished essential data or reagents

### Author ORCIDs

Michael X Cohen, http://orcid.org/0000-0002-1879-3593

## Additional files

### Major datasets

The following previously published datasets were used:

| Author(s) | Year | Dataset title | Dataset URL | Database, license, and accessibility information |
|---|---|---|---|---|
| Mizuseki K, Sirota A, Pastalkova E, Diba K, Buzsáki G | 2016 | hc-3 | https://crcns.org/data-sets/hc/hc-3/about-hc-3 | Publicly available at CRCNS - Collaborative Research in Computational Neuroscience |
| Fujisawa S, Amara-singham A, Harri-son MT, Peyrache A, Buzsáki G | 2016 | pfc-2 | https://crcns.org/data-sets/pfc/pfc-2/about-pfc-2 | Publicly available at CRCNS - Collaborative Research in Computational Neuroscience |

| Essen DV, Ugurbil K | 2016 | MEG resting-state | https://db.humancon-nectome.org/app/ac-tion/DisplayItemAction/search_value/Connecto-meDB_E10375/search_element/xnat:megSes-sionData/search_field/xnat:megSessionData.ID/ | Publicly available at Human Connectome (http://www.humanconnectome.org/about/project/resting-MEG.html) |

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
