## [Decision Letter]

Thank you for submitting your article "Multivariate cross-frequency coupling via generalized eigendecomposition" for consideration by *eLife*. Your article has been reviewed by three peer reviewers, and the evaluation has been overseen by a Reviewing Editor and David Van Essen as the Senior Editor. The following individual involved in review of your submission has agreed to reveal his identity: Christopher J Honey (Reviewer #2).

The reviewers have discussed the reviews with one another and the Reviewing Editor has drafted this decision to help you prepare a revised submission.

The paper nicely addresses an alternative method to measuring cross-frequency coupling, namely using a multivariate-components-based framework. The paper notes a number of challenges for existing analytic approaches and proposes and examines a new analytic framework named generalized eigendecomposition CFC (gedCFC).

The reviewers all agreed that the basics of the paper are novel, important and sound. However, they also all agreed that the paper was in need of an extensive rewriting coupled with some additional analyses to bring this to a level appropriate for *eLife* readership. Although we typically at *eLife* will provide a condensed list of essential revisions, in this case, I am appending the actual reviews below because they all nicely describe the needed changes.

*Reviewer #1:*

This paper reports results of various methods of applying generalized eigendecomposition to perform cross-frequency correlation analysis. The discussion in the Introduction seems to be too cursory to be of much benefit to those not already familiar with the idea, so depending on the purpose and target audience, some elaboration may be warranted.

There seems to be an error in the statement of the equation: SW = LRW (where L stands for Λ) – I think SW = RWL is meant.

I would humbly suggest giving an intuitive illustrative example in the Introduction to demonstrate the "maximization and minimization" effect referred to, perhaps using the Rayleigh coefficient framework.

The Discussion seems to be somewhat lacking in structure. It may be helpful to move some of the discussion in the corresponding Methods sections to the general Discussion section. The identification of different methods is interesting, but it would be nice to have a prior list of "general CFC contexts" in the Introduction, in which one would like to apply the GED framework, perhaps with simple examples of each case, then the methods description referring to the corresponding contexts, focusing on detailed description of the simulation/data used and method employed, followed by a more general discussion in the Discussion section.

I believe the paper could benefit from a quantitative report of the resulting covariances obtained with the covariance matrices, e.g. the theta and gamma template vectors with each of the two covariance matrices used in the method, as well as the generalized eigenvalue. And it would be of use to know the relative power in the high and low frequency bands used in the simulations, with some corresponding discussion of minimal relative power necessary e.g. gamma, to obtain interpretable results, and the applicability of the required power levels assumptions to real data, e.g. EEG.

The paper is likely to be useful for specialists working in the area, but if a wider audience is intended, a more generous introductory section and structured Discussion section would be helpful.

*Reviewer #2:*

The goal of this paper is "to highlight that significant insights into the neural mechanisms and implications of CFC can be gained by expanding the repertoire of data analyses, as well as theoretical conceptualizations, from a single-electrode-sinusoid-based framework to a multivariate-components-based framework." The author notes a number of challenges for existing CFC approaches include non-stationarities and sharp transients in signal, non-uniform phase distributions, and the fact that CFC sources and targets are spatially distributed and potentially overlapping. To address these challenges they propose a new analytic framework named generalized eigendecomposition CFC (gedCFC). Although the method proposed is very promising, the manuscript reads more like a proof of principle than a convincing demonstration. This is not because the framework is flawed, but because the range of applications discussed (5 different settings) leads to a lack of focus, and so the strengths and weaknesses of the framework are not clearly enough tested and reported within each setting.

1) The paper describes five methods for characterizing changes in spatiotemporal properties of brain data, using a common linear algebraic formalism. On the one hand, the range of methods is valuable because CFC measures are applied in a range of contexts to address a range of hypotheses. However, this also means that manuscript lacks conceptual unity and focus. For example, it is not entirely clear which of the 5 methods are solutions to which of the problems (e.g. non-stationarities, sharp transients, non-uniform phase distributions, etc.) with existing CFC methods. I would recommend:i) try to group the 5 methods into clusters with a common purpose (for example Methods 1 and 2 could be grouped), rather than simply using numeric labels;ii) assign names to the different approaches;iii) create a table which summarizes, for each method, the kind of hypothesis it is best suited for testing, which problems it solves, and which problems it does not solve;iv) for each of the claims above concerning the problems solved by gedCFC, please design a test that specifically demonstrates that claim. The manuscript already includes valuable demonstration examples that do perform this function, but it is not always explicitly specified which facet of gedCFC they are meant to test.

More generally, I found that the readability of the manuscript decreased from Method 3 onward, as more complex methods were introduced. In method 4 delay-coordinate embedding is introduced, just in passing, but this is a complicated and rich area of research. The investigation of how to choose the correct parametrization (e.g. number of time delays) in Method 4 could be a paper of its own. So, again, it is nice to see how the gedCFC framework can be applied, but the range of uses also limits the manuscript's depth and clarity.

2) Why compare gedCFC against methods, such as PACz, which are (according to the citations in the introduction) known to be flawed? The manuscript would be more compelling if gedCFC was compared against state-of-the-art methods (some of which are mentioned in the Introduction) and/or against the most successful of the methods reviewed in Tort et al. (2010, J Neurophys). In addition, when existing methods fail, it would help to specify precisely why they are failing. For example, it could be that gedCFC has, effectively, greater statistical power when applied to multi-channel data due to pooling of information across multiple electrodes, but it was unclear to me whether this was the core reason for its advantage over PACz.

3) "This alleviates the feature of principle components analysis that makes it suboptimal for brain data: components are forced to be orthogonal but neural dynamics (certainly at the meso- to macroscopic scale of LFP and EEG) are not orthogonal."

It seems unlikely to me that this undesirable feature of PCA – the fact that, e.g., the second eigenvector's direction is constrained by the direction of the first eigenvector – is truly absent from the gedCFC framework. Generally, there is a distinction drawn between (i) dimensionality reduction methods, such as PCA, which rotate data so that it can be compactly re-expressed; and (ii) source separation methods, such as ICA, which aim to identify statistically distinct generators within a mixed signal. Although in many cases PCA does approximate the function of source separation, this is not its goal, and the orthogonality assumption is a testament to the fact that compact expression is prioritized over source separation.

My (admittedly intuitive) impression is that the gedCFC framework is closer to a dimensionality reduction method than to a source separation method. Specifically, even though the components in gedCFC are not, in general, constrained to be orthogonal, this does not imply that (e.g.) the direction of the eigenvector of second-largest eigenvalue is unaffected by the direction of the eigenvector whose eigenvalue is largest. Consider the special case in which R^(-1)S is indeed symmetric: in this case, the eigenvectors are again constrained to be orthogonal and the well-known PCA problems return. So, should we expect PCA-like difficulties whenever R^(-1)S is near-symmetric? What is our guarantee that these issues do not arise even when R^(-1)S is far from symmetric?

The manuscript already presents some examples of cases where gedCFC successfully extracts signals from a mixture, but in these cases the mixed signals have different frequencies, so the filtering operations may be doing much of the work. To demonstrate that gedCFC really does not run into the interpretational difficulties seen with PCA, I recommend setting up a situation in which multiple signals (say 3) with similar spatial profiles, similar frequencies, and overlapping functional response profiles, are mixed – a case in which PCA should fail – and then show that gedCFC does not suffer from similar problems.

4) Because the model works by comparing a reference covariance to a target covariance, it assumes a binary transition between two modes. But how well would the model perform if the underlying generative process is actually changing continuously? For example if the generative model of gamma power is something like probability(gamma burst) ~ phase(theta), then a method that assumes a continuous variation as a function of phase will have far greater statistical power than a method which simply contrasts peak and off-peak data? Now, Method 3 may present an answer this "binary assumption" issue, because a continuous time-varying signal is convolved and included the model. But in this case, does not one not re-encounter the problem of non-stationarity and sharp transients, because now the estimation of the low-frequency envelope matrix, B, could be affected by these factors? Similar to the point I made in (1) above, I would like to see a more careful discussion of the strengths and weaknesses of each method, so that the manuscript explicitly describes for each method, which of the common challenges for CFC are overcome, and which are not overcome.

5) The paper would be more practically useful if it provided a slightly more thorough treatment of at least one of the statistical methods for testing components, and noted briefly any problems that may arise in this setting.

Reviewer #3:

This paper describes a novel technique to calculate cross-frequency coupling using generalized eigenvalue decomposition. The technique is novel and has several significant advantages that have currently not been clearly addressed, such as its independence of selecting electrodes and not assuming a sinusoidal shape. Moreover, the technique can be used in several ways (5 methods are described in the paper). The paper is well written; as I was reading it and wrote down questions, a subsequent section would often already provide an answer. Overall, I think the paper is of high quality and I only have a few comments that can be easily addressed.

1) It is not quite clear to me whether the method assumes that multiple electrodes have the exact same phase for an underlying rhythm. For example, if the method were to be used to analyze LFP recordings across motor cortex, with traveling β oscillations, where there is a shift in phase across electrodes, how would that reflect upon component waves?

2) It is said that the quality of gedCFC components is related to the quality of the covariance matrices, and a sufficient amount of clean data ensures high-quality covariance matrices. Can this be quantified: is there a way to compute from one of the data sets used in the paper how the amount of data used relates to the quality of the gedCFC components?

3) In method 3 example recordings from a human epilepsy patient are used with recordings of the medial temporal lobe and electrode Cz. Why would there be any coupling in these data? How does referencing affect the method? More anterior temporal lobe channels are also closer the face with potential muscle artifacts, which may not even be removed with bipolar referencing.

---

## [Author Response]

*[…] Reviewer #1:*

*This paper reports results of various methods of applying generalized eigendecomposition to perform cross-frequency correlation analysis. The discussion in the Introduction seems to be too cursory to be of much benefit to those not already familiar with the idea, so depending on the purpose and target audience, some elaboration may be warranted.*

*There seems to be an error in the statement of the equation: SW = LRW (where L stands for Λ) – I think SW = RWL is meant.*

Indeed, that was a typo (copied from the initial introduction of a single eigenvalue). Thanks for catching that.

*I would humbly suggest giving an intuitive illustrative example in the Introduction to demonstrate the "maximization and minimization" effect referred to, perhaps using the Rayleigh coefficient framework.*

I agree that an intuitive example would be useful. However, I would guess that anyone who is familiar with the Rayleigh quotient would also understand GED. I’ve opted for a visual example presented in Figure 1—figure supplement 1, (inspired by Figure 6 in Blankertz et al., 2008) that I hope is more intuitive to readers without a strong mathematical background.

Furthermore, I added several new supplemental figures that provide graphical illustrations of the methods.

*The Discussion seems to be somewhat lacking in structure. It may be helpful to move some of the discussion in the corresponding Methods sections to the general Discussion section. The identification of different methods is interesting, but it would be nice to have a prior list of "general CFC contexts" in the Introduction, in which one would like to apply the GED framework, perhaps with simple examples of each case, then the methods description referring to the corresponding contexts, focusing on detailed description of the simulation/data used and method employed, followed by a more general discussion in the Discussion section.*

I have tried to improve the readability and organization of the manuscript, including more discussions of the assumptions and goals of each specific method. The Discussion section is now more focused, and several method-specific points have been incorporated into the relevant Results section. Furthermore, the revision includes a table that summarizes the key goals and assumptions of each method.

*I believe the paper could benefit from a quantitative report of the resulting covariances obtained with the covariance matrices, e.g. the theta and gamma template vectors with each of the two covariance matrices used in the method, as well as the generalized eigenvalue.*

I’m afraid I don’t completely understand this request. The covariance matrices themselves are not terribly insightful and they are dependent on the location of the simulated dipole, which is arbitrary and not relevant for the method. If the reviewer was referring to the use of GED to obtain within-frequency components (e.g., the theta component to extract theta phase), then this is a validated method in neuroscience, and I have added additional references in the subsection “Using GED to identify the low-frequency rhythm” for interested readers. If the reviewer is referring to assessing the quality of a covariance matrix, this is now discussed in the last paragraph of the subsection “Nonstationarities and condition differences”. If I have misunderstood this comment, then I apologize and ask the reviewer to clarify.

*And it would be of use to know the relative power in the high and low frequency bands used in the simulations, with some corresponding discussion of minimal relative power necessary e.g. gamma, to obtain interpretable results, and the applicability of the required power levels assumptions to real data, e.g. EEG.*

The relative power between high and low frequency bands is not important; indeed, they are not directly compared. Instead, it is only the covariance matrices within a frequency band from different time periods that are compared (or, in other cases, relative to the broadband power). For example, in the simulations shown in Figure 1, the gamma peak is present but weaker than the “distractor” (non-coupled) gamma; in the simulations shown in Figure 3 and Figure 5, the gamma peaks are hardly visible in the power spectrum. This is now discussed in the Results section among other places.

*The paper is likely to be useful for specialists working in the area, but if a wider audience is intended, a more generous introductory section and structured Discussion section would be helpful.*

I hope the reviewer agrees that the revised version is more approachable. I welcome any additional specific suggestions for improvement.

*Reviewer #2:*

*The goal of this paper is "to highlight that significant insights into the neural mechanisms and implications of CFC can be gained by expanding the repertoire of data analyses, as well as theoretical conceptualizations, from a single-electrode-sinusoid-based framework to a multivariate-components-based framework." The author notes a number of challenges for existing CFC approaches include non-stationarities and sharp transients in signal, non-uniform phase distributions, and the fact that CFC sources and targets are spatially distributed and potentially overlapping. To address these challenges they propose a new analytic framework named generalized eigendecomposition CFC (gedCFC). Although the method proposed is very promising, the manuscript reads more like a proof of principle than a convincing demonstration. This is not because the framework is flawed, but because the range of applications discussed (5 different settings) leads to a lack of focus, and so the strengths and weaknesses of the framework are not clearly enough tested and reported within each setting.*

Thank you for your evaluation and for your many comments. I agree with this assessment, but I would not consider it to be a weakness. The intention was not to propose a single method with a single set of parameters; instead, the goal was to provide the CFC field with an alternative perspective for thinking about and analyzing CFC in multichannel data. The specific implementation of the gedCFC framework should be left to the researcher, depending on the data and goals of the analysis. I agree that a paper presenting only one method would be more focused as a piece of written literature, but it would also be less useful as a scientific contribution.

Furthermore, because the different methods are generally similar to each other, I do not think it is necessary to provide a full complement of testing for each method; this was the reason for setting up the different simulations in different ways. I have tried to make the consistency across methods more apparent in the revised version.

As detailed below, many aspects of the revision have been clarified and tightened. I hope the reviewer agrees.

*1) The paper describes five methods for characterizing changes in spatiotemporal properties of brain data, using a common linear algebraic formalism. On the one hand, the range of methods is valuable because CFC measures are applied in a range of contexts to address a range of hypotheses. However, this also means that manuscript lacks conceptual unity and focus. For example, it is not entirely clear which of the 5 methods are solutions to which of the problems (e.g. non-stationarities, sharp transients, non-uniform phase distributions, etc.) with existing CFC methods. I would recommend:i) try to group the 5 methods into clusters with a common purpose (for example Methods 1 and 2 could be grouped), rather than simply using numeric labels;ii) assign names to the different approaches;iii) create a table which summarizes, for each method, the kind of hypothesis it is best suited for testing, which problems it solves, and which problems it does not solve;iv) for each of the claims above concerning the problems solved by gedCFC, please design a test that specifically demonstrates that claim. The manuscript already includes valuable demonstration examples that do perform this function, but it is not always explicitly specified which facet of gedCFC they are meant to test.*

Thank you for these suggestions. The summary table is a great idea, and is included in the revised version. I decided against explicit grouping of methods, as they do not clearly group together in a parsimonious way. The text now discusses better how the different Methods are related to each other. I also decided against giving each method a name, as the names would either be (1) as long as their descriptions or (2) difficult to remember.

I appreciate that the reviewer feels the paper “lacks conceptual unity”; however, the goal here is not to present a simple literary work, but instead to present a group of related data analysis techniques, linked by a single mathematical and conceptual framework, that will benefit the electrophysiology community. Readers will be able to determine the most appropriate method for their data and hypotheses, rather than relying on (and, typically, being disappointed by) one particular black-box algorithm that may or may not be appropriate for their data.

The revision is more explicit about the underlying assumptions and limitations of each method, including extra text at the beginning and end of each of the subsections that introduce each method.

It is not really the case that there is a one-to-one mapping between a CFC problem and a gedCFC method. All of the methods address the limitations faced by standard CFC analyses. But the purpose of gedCFC is not simply that it addresses limitations of existing methods; it is a totally new technique for measuring CFC. Therefore, it is not so simple as writing “method A has problem X, and gedCFC-method-Y resolves it.” That would be analogous to saying that the primary purpose of EEG is that fMRI has poor temporal resolution.

*More generally, I found that the readability of the manuscript decreased from Method 3 onward, as more complex methods were introduced. In method 4 delay-coordinate embedding is introduced, just in passing, but this is a complicated and rich area of research. The investigation of how to choose the correct parametrization (e.g. number of time delays) in Method 4 could be a paper of its own. So, again, it is nice to see how the gedCFC framework can be applied, but the range of uses also limits the manuscript's depth and clarity.*

Indeed, parameterization of delay-embedded matrices is a non-trivial problem. But without making this paper a full-length book, it is not possible to go into great detail about each method. I appreciate that this necessarily requires sacrificing some focus, but as written above, the goal here was to present a framework, not a black-box recipe for a specific analysis.

I have tried to make the paper more readable. However, I believe it simply will necessarily be the case that people will need some background in signal processing and matrix analysis to be able to apply the analyses described here.

More concretely, the revision includes more approachable introductions to each method, and pictorial overviews of the mechanics of each method.

*2) Why compare gedCFC against methods, such as PACz, which are (according to the citations in the introduction) known to be flawed? The manuscript would be more compelling if gedCFC was compared against state-of-the-art methods (some of which are mentioned in the Introduction) and/or against the most successful of the methods reviewed in Tort et al. (2010, J Neurophys). In addition, when existing methods fail, it would help to specify precisely why they are failing. For example, it could be that gedCFC has, effectively, greater statistical power when applied to multi-channel data due to pooling of information across multiple electrodes, but it was unclear to me whether this was the core reason for its advantage over PACz.*

PACz is a standard and widely used measure of CFC, probably the most commonly used method. It has its limitations, but I would not consider it to be “flawed.” Instead, it can overestimate or underestimate true PAC in the case of non-sinusoidal oscillations, depending on the nature of the non-stationarities (this is detailed in van Driel et al., 2014, J Neuro Methods). PACz will produce the same results as the distribution analysis described in Tort et al. 2010 when power has a monopolar distribution over low-frequency phase (this is what is observed empirically) and when the lower frequency phase angles are uniformly distributed.

The reviewer is correct that the increased SNR of gedCFC is partly due to the weighted average of electrodes. However, the primary benefit is the robustness to nonstationarities in the low-frequency and the high-frequency dynamics, which are the primary sources of bias in PACz, as described in the manuscript. These points are now detailed in the paper.

*3) "This alleviates the feature of principle components analysis that makes it suboptimal for brain data: components are forced to be orthogonal but neural dynamics (certainly at the meso- to macroscopic scale of LFP and EEG) are not orthogonal."*

*It seems unlikely to me that this undesirable feature of PCA – the fact that, e.g., the second eigenvector's direction is constrained by the direction of the first eigenvector – is truly absent from the gedCFC framework. Generally, there is a distinction drawn between (i) dimensionality reduction methods, such as PCA, which rotate data so that it can be compactly re-expressed; and (ii) source separation methods, such as ICA, which aim to identify statistically distinct generators within a mixed signal. Although in many cases PCA does approximate the function of source separation, this is not its goal, and the orthogonality assumption is a testament to the fact that compact expression is prioritized over source separation.*

*My (admittedly intuitive) impression is that the gedCFC framework is closer to a dimensionality reduction method than to a source separation method. Specifically, even though the components in gedCFC are not, in general, constrained to be orthogonal, this does not imply that (e.g.) the direction of the eigenvector of second-largest eigenvalue is unaffected by the direction of the eigenvector whose eigenvalue is largest. Consider the special case in which R^(-1)S is indeed symmetric: in this case, the eigenvectors are again constrained to be orthogonal and the well-known PCA problems return. So, should we expect PCA-like difficulties whenever R^(-1)S is near-symmetric? What is our guarantee that these issues do not arise even when R^(-1)S is far from symmetric?*

*The manuscript already presents some examples of cases where gedCFC successfully extracts signals from a mixture, but in these cases the mixed signals have different frequencies, so the filtering operations may be doing much of the work. To demonstrate that gedCFC really does not run into the interpretational difficulties seen with PCA, I recommend setting up a situation in which multiple signals (say 3) with similar spatial profiles, similar frequencies, and overlapping functional response profiles, are mixed – a case in which PCA should fail – and then show that gedCFC does not suffer from similar problems.*

GED is often referred to as a source separation method, both in the EEG (e.g., Parra 2003) and in the mathematics and engineering literatures (e.g., Tomé 2006). It is also, as the reviewer suggests, referred to as a dimensionality reduction method, although source separation is also dimensionality reduction (the reverse, of course, is not true).

Although the paper provides only a theoretical remark about the superiority of GED over PCA, this has been empirically demonstrated in several papers in neuroscience, a few of which are now cited in the text. Given the many instances in the literature showing the superiority of GED over PCA for source separation, and given that PCA does not fit into the gedCFC framework, and that the purpose here is to provide a method for CFC and PCA is not a method for CFC, it seems redundant and beyond the scope of this paper to provide another formal demonstration of GED vs. PCA for source separation. PCA was mentioned only to introduce readers to GED, because in my experience, most electrophysiologists are familiar with PCA and this provides an easy scaffold for understanding GED. I believe the new Figure 1—figure supplement 1 also helps clarify the improvement of GED over PCA for source separation.

Furthermore, I’m not sure I follow the reviewer’s suggestion: If three signals have the same spatial and spectral profiles and overlapping functional response profiles, then it would seem to me that these are actually reflecting a single functional source. gedCFC is not a dipole-localization technique; in fact, it works entirely indifferently to spatial configuration (as illustrated in Figure 1—figure supplement 2). The term “source” has varied interpretations in neuroscience; components analyses refer to statistical sources, as opposed to anatomical sources (the two sometimes overlap but often do not).

As for temporal filtering, the reviewer’s point about temporal filtering taking care of the separation is not correct. For one thing, many of the analyses reported here did not involve any temporal filtering to obtain the CFC components. Secondly, no spurious CFC was identified for strong gamma components that were in the data but not locked to the low-frequency rhythm.

Regarding the symmetry: It is difficult to imagine a situation in which R^-1^S will just happen to be symmetric. Symmetric matrices are very rare, and basically only occur by specific construction (multiplying a matrix by its transpose). Even multiplying a symmetric matrix with the identity matrix that has a single off-diagonal element replaced by .1 will produce a non-symmetric matrix. Furthermore, the law (it is mathematically required; it is not an assumption) that a symmetric matrix has orthogonal eigenvectors simply states that if the eigenvalues are different, the eigenvectors are orthogonal (if there are repeated eigenvalues, the vectors need not be orthogonal, but most algorithms select them to be orthogonal); there is nothing in the math about vectors approaching orthogonality as a matrix approaches symmetry.

Finally, regarding the point about the direction of the second vector being constrained by the direction of the first vector, this is true, but it is also true for ICA and any other source separation method, in the sense that two eigenvectors cannot point in exactly the same direction (analogously, two ICs cannot be exactly the same). For eigendecomposition, any two distinct eigenvalues will produce different eigenvectors. Therefore, any one eigenvector will constrain the direction of any other eigenvector in the senses that (1) their angles are nonzero for distinct eigenvalues and (2) they are orthogonal for symmetric matrices. Even for reduced-rank matrices that lack a full spectrum of eigenvalues, the uncertainty about in-plane eigenvectors affects only the smaller components that are not considered in gedCFC.

I’m not sure what to do about this comment. It is beyond the scope of this paper to provide a lengthy discussion about linear algebra concepts (readers can find such information elsewhere). I hope the supplemental figure illustrating GED in a 2D example is informative.

*4) Because the model works by comparing a reference covariance to a target covariance, it assumes a binary transition between two modes. But how well would the model perform if the underlying generative process is actually changing continuously? For example if the generative model of gamma power is something like probability(gamma burst) ~ phase(theta), then a method that assumes a continuous variation as a function of phase will have far greater statistical power than a method which simply contrasts peak and off-peak data? Now, Method 3 may present an answer this "binary assumption" issue, because a continuous time-varying signal is convolved and included the model. But in this case, does not one not re-encounter the problem of non-stationarity and sharp transients, because now the estimation of the low-frequency envelope matrix, B, could be affected by these factors? Similar to the point I made in (1) above, I would like to see a more careful discussion of the strengths and weaknesses of each method, so that the manuscript explicitly describes for each method, which of the common challenges for CFC are overcome, and which are not overcome.*

We should distinguish between two different stages of analysis: (1) designing the spatial filter; (2) using the spatial filter to extract a component that is then used for analysis. Only the first stage involves a binary discretization; the second stage provides a continuous measure of activity.

In fact, the situation the reviewer describes (gamma power varying continuously as a function of theta phase) is exactly what was implemented in the simulations for Methods 1 and 2. As shown in Figure 1 and Figure 3, the methods work quite well at recovering continuous gamma power as a function of continuous theta phase, even though the spatial filter was defined only using two covariance matrices taken from discrete time windows. As long as the spatial structure of the network remains the same, the method will work. In this sense, the reviewer is not correct that the discretization is a weakness. This distinction is now clarified on “Generalized eigendecomposition” and in Figure 3 legend.

With regards to the final point, the revised version now more clearly states the assumptions, advantages, goals, and limitations of each method. This information is listed at the start and end of each subsection of the Results.

*5) The paper would be more practically useful if it provided a slightly more thorough treatment of at least one of the statistical methods for testing components, and noted briefly any problems that may arise in this setting.*

Statistical evaluations are now presented in two figures, and the section about statistical evaluation is expanded.

Reviewer #3:

*This paper describes a novel technique to calculate cross-frequency coupling using generalized eigenvalue decomposition. The technique is novel and has several significant advantages that have currently not been clearly addressed, such as its independence of selecting electrodes and not assuming a sinusoidal shape. Moreover, the technique can be used in several ways (5 methods are described in the paper). The paper is well written; as I was reading it and wrote down questions, a subsequent section would often already provide an answer. Overall, I think the paper is of high quality and I only have a few comments that can be easily addressed.*

*1) It is not quite clear to me whether the method assumes that multiple electrodes have the exact same phase for an underlying rhythm. For example, if the method were to be used to analyze LFP recordings across motor cortex, with traveling β oscillations, where there is a shift in phase across electrodes, how would that reflect upon component waves?*

As long as the phase lags are consistent over repeated epochs, they are not detrimental to the analysis. This is now in the first paragraph of the subsection “Nonstationarities and condition differences”. Method 4 would reveal phase lags as delays in the filter time series for different channels, e.g., a slanted bar in Figure 7. I have seen this in other datasets, but not in the data shown here.

*2) It is said that the quality of gedCFC components is related to the quality of the covariance matrices, and a sufficient amount of clean data ensures high-quality covariance matrices. Can this be quantified: is there a way to compute from one of the data sets used in the paper how the amount of data used relates to the quality of the gedCFC components?*

This is an important point, and it was too ambiguous in the original submission. Unfortunately, there are no formal criteria for what makes a covariance matrix good enough for eigendecomposition. This is an important issue and it is now discussed in more detail in the paper (subsection “Nonstationarities and condition differences”, third paragraph).

*3) In method 3 example recordings from a human epilepsy patient are used with recordings of the medial temporal lobe and electrode Cz. Why would there be any coupling in these data? How does referencing affect the method? More anterior temporal lobe channels are also closer the face with potential muscle artifacts, which may not even be removed with bipolar referencing.*

There is considerable evidence for the role of hippocampal-prefrontal interactions in memory consolidation and retrieval. This has been studied in rodents using electrophysiology and lesions, and in humans using invasive electrophysiology, MEG, and fMRI. A popular review paper is now cited in the last paragraph of the subsection “Method 3: Low-frequency waveform shape as a bias filter on sphered data”.

The hippocampal electrode was locally referenced to its average while Cz was referenced to linked mastoids. Intracranial sources, particularly with local referencing, are not affected by facial muscles. Nonetheless, this N=1 dataset was included as proof-of-principle illustration in empirical data; strong claims about brain function are not made based on these data.